# Filament formation drives catalysis by glutaminase enzymes important in cancer progression

Shi Feng [1], Cody Aplin [1], Thuy-Tien T. Nguyen [1], Shawn K. Milano[1] & Richard A. Cerione [1,2] ✉

The glutaminase enzymes GAC and GLS2 catalyze the hydrolysis of glutamine to glutamate, satisfying the 'glutamine addiction' of cancer cells. They are the targets of anti-cancer drugs; however, their mechanisms of activation and catalytic activity have been unclear. Here we demonstrate that the ability of GAC and GLS2 to form filaments is directly coupled to their catalytic activity and present their cryo-EM structures which provide a view of the conformational states essential for catalysis. Filament formation guides an 'activation loop' to assume a specific conformation that works together with a 'lid' to close over the active site and position glutamine for nucleophilic attack by an essential serine. Our findings highlight how ankyrin repeats on GLS2 regulate enzymatic activity, while allosteric activators stabilize, and clinically relevant inhibitors block, filament formation that enables glutaminases to catalyze glutaminolysis and support cancer progression.

Metabolic reprogramming is well-established as a hallmark of cancer progression[1–3]. A shift in the metabolic requirements of cancer cells compared to normal cells, known as the Warburg effect[4–6], results in their dependence on glutamine as an energy source in addition to glucose. Glutamine catabolism, which starts with the hydrolysis of glutamine to glutamate catalyzed by the glutaminase family of enzymes, becomes highly upregulated in cancer cells to compensate for the uncoupling of the glycolytic pathway from the TCA cycle due to the Warburg effect, thus providing the biosynthetic precursors necessary for their enhanced proliferation and survival[7–10]. Given the resulting glutamine addiction of cancer cells, the development of allosteric small molecule inhibitors that target glutaminase activity is being examined as a potential therapeutic strategy to block tumor progression in various malignancies including aggressive breast, lung, and pancreatic cancers, as well as for their effectiveness when combined with other anti-cancer drugs[11–16]. The glutaminases are encoded by two independent genes, *gls* and *gls2*. *Gls* encodes kidney-type glutaminase (KGA) and its C-terminal truncated splice variant glutaminase C (GAC) which is highly expressed in many cancers. *Gls2* encodes only one active isozyme, liver-type glutaminase (LGA; from hereon

designated as GLS2). While GLS2 has been suggested to serve as a tumor suppressor by directly binding the Rac GTPase and promoting ferroptosis in hepatocellular carcinoma[17,18], it has also been shown to be necessary for cancer progression as we and others have found that it is essential for the growth and survival of luminal subtype receptor-positive breast cancer cells[9,19]. Moreover, GLS2 is highly expressed in ovarian, lung, and colorectal cancers and has been suggested to compensate for GAC downregulation or inhibition[10,20].

All glutaminase enzymes share a catalytic domain containing identical active site residues and a conserved mechanism for hydrolyzing glutamine to glutamate[21–23]. The catalytic efficiency of the glutaminase enzymes is markedly increased in the presence of inorganic phosphate in vitro[24,25], although the concentration of this allosteric activator required to promote their enzymatic activity may not be attainable in many physiological contexts suggesting that other anionic activators likely exist. Understanding more about the catalytic mechanism used by these enzymes and the structural changes that they need to undergo to become activated will be important for further developing drug candidates that effectively block their enzymatic activity. Potent inhibitors targeting GAC are available, the most

[1]Department of Chemistry and Chemical Biology, Cornell University, Ithaca, NY 14853, USA. [2]Department of Molecular Medicine, Cornell University, Ithaca, NY 14853, USA. ✉e-mail: rac1@cornell.edu

effective being the BPTES/CB-839 group of compounds with CB-839 and others such as UPGL004 having nanomolar binding affinity[26,27]. However, the development of specific and potent allosteric inhibitors of GLS2 has not been as successful, as currently the best examples are the 968-class of molecules[9,15] which have binding affinities for the enzyme only in the micromolar range.

It has been known for some time that a number of metabolic enzymes assemble into filament-like structures which may play a key regulatory role in their catalytic activity[28]. Recently it has been shown that GAC can assemble into long filaments in cells to facilitate efficient glutamine consumption[29]. While the formation of these oligomeric supramolecular assemblies has been suggested to be associated with the in vitro activation of GAC by inorganic phosphate[30,31], an essential question concerns if the assembly of such higher-order oligomers is in fact directly coupled to catalytic activity. Moreover, it has yet to be established whether filament formation is a common mechanism for all glutaminase isozymes. This is an especially important question for GLS2 given that it might be differentially regulated due to the presence of ankyrin repeats, which are absent on the C-terminal truncated splice variant GAC[32,33], and because it can sometimes exert the opposite functional effects from GAC on cancer progression.

In this study, we show that GAC as well as GLS2 undergoes filament formation in the presence of the substrate glutamine and the activator inorganic phosphate, with the assembly of these higher-order oligomers being directly coupled to glutamine hydrolysis and product (glutamate) release. We further present the high-resolution structures for a GAC filament bound to its anionic activator inorganic phosphate (Pi), and for a constitutively active GLS2 filament, determined by cryo-electron microscopy (cryo-EM). These filamentous structures now provide us with several important insights into how key regions of the glutaminases and their essential residues are positioned to achieve maximal catalytic activity, as well as shed light on how anionic activators, allosteric inhibitors, and ankyrin repeats can impart their regulatory effects on enzymes that play critical roles in cancer progression.

## Results

### General structural features of the glutaminase enzymes

Several high-resolution X-ray structures for KGA and GAC have been reported; however, thus far, there has not been a structural determination for the full-length apo-GLS2 protein. Therefore, we obtained a cryo-EM structure for wild-type, full-length GLS2. Figure 1a, b show the

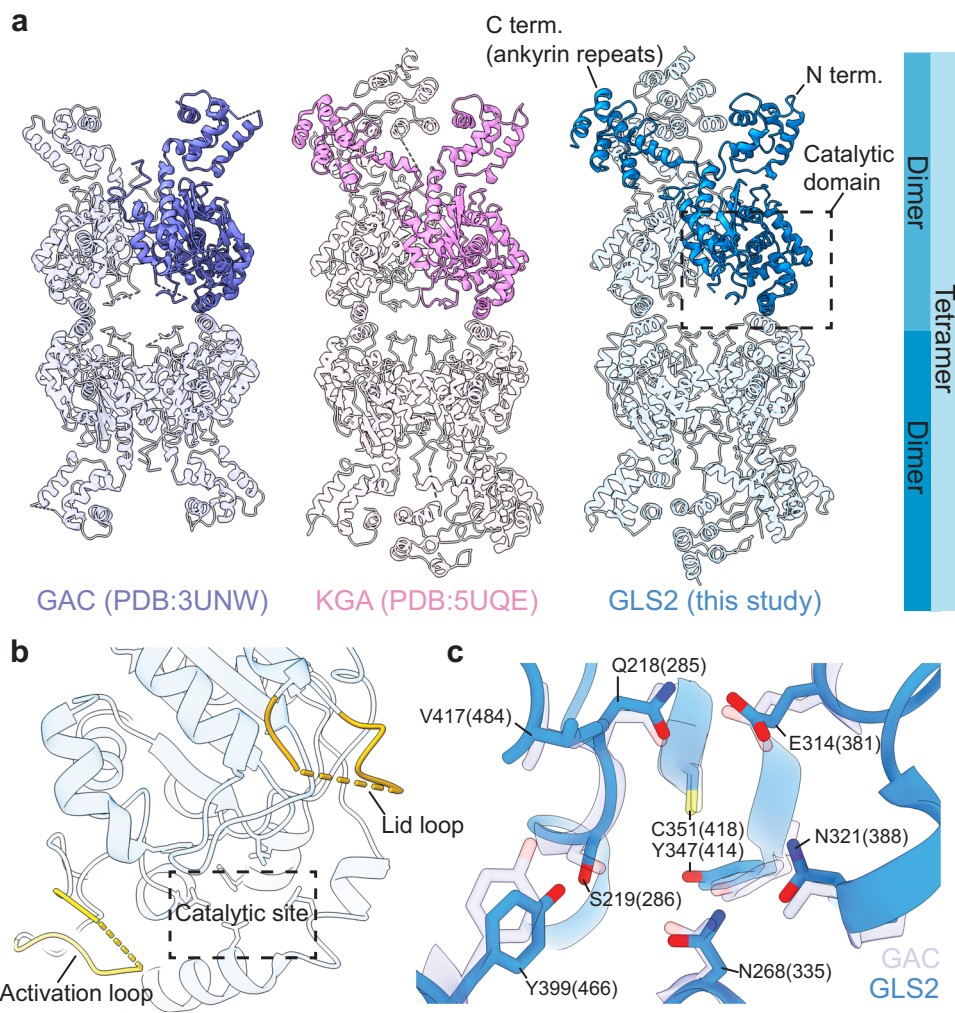

**Fig. 1 | The cryo-EM structure of human full-length apo-GLS2 and its comparison with the X-ray crystal structures of the GLS isoforms. a** The structures of tetrameric GAC, KGA, and GLS2 (from left to right). One monomer in each tetramer is highlighted in solid color. The glutaminase catalytic domain (dashed box in (**a**)). **b** Zoomed in view of the catalytic domain showing the activation loop (light yellow), the catalytic site (dashed box), and the lid loop (dark yellow). **c** The conserved catalytic residues of GLS2 (blue) and GAC (PDB ID: 3UNW, light purple). Residues are numbered according to human GLS2 and GAC numbering is shown in parentheses.

3.1 Å structure of the human apo-GLS2 tetramer and the relative positions of the key regions for catalytic activity (i.e., the 'activation loop', the catalytic site, and the 'lid loop' that opens and closes over the catalytic domain)[23]. Figure 1a also highlights the three major domains of a monomeric unit of GLS2, namely, the N-terminal region, C-terminal ankyrin repeats, and the highly conserved catalytic site. The overall structure of the apo-GLS2 tetramer strongly resembles those of the apo-forms of GAC and KGA solved by X-ray crystallography (Fig. 1a)[24,33], and the active sites within each GLS2 subunit show the same general features as a previously reported crystal structure of the enzyme's limit catalytic domain as well as those of GAC[22] (Fig. 1c). The fact that these structural features are highly conserved among the glutaminase enzymes indicate that they share a common catalytic mechanism. We analyzed several key structural elements in the catalytic domain of GLS2 that have been suggested to exert an allosteric regulation of glutaminase activity. The stretch of residues Gly248-Glu258, which has been shown to play an essential role in catalysis and thus referred to as the 'activation loop'[23,34], is disordered and flexible in the GLS2 structure (Fig. 1b). In GAC, this loop contains the binding site for the BPTES-class of allosteric inhibitors which includes CB-839, a drug candidate that has been in clinical trials[26]. The loop forms a stable, inactive conformation when bound to BPTES and to its more potent analogs, CB-839 (not shown) and UPGL0004[35,36] (Supplementary Fig. 1). However, the activation loop of GLS2 differs from KGA and GAC by two residues, rendering GLS2 insensitive to these inhibitors[24]. In addition, residues Tyr182-Lys188 in GLS2, which correspond to a loop in KGA and GAC forming a 'lid' that needs to close over bound substrates for catalysis to occur, are also flexible like the activation loops in the GLS2 structure. This is in contrast to the previously reported crystal structure of the GLS2 limit catalytic domain, which showed the lid region adopting an open conformation[22].

## Glutaminase filament formation is directly coupled to catalytic activity

The glutaminase enzymes have been shown to undergo dimer-to-tetramer transitions upon binding activators such as inorganic phosphate (Pi)[25,37], as a necessary step for enzymatic activity. However, high-resolution X-ray crystal structures of the tetrameric forms of glutaminase enzymes have failed to shed light on the critical changes in their catalytic sites that promote the hydrolysis of glutamine to glutamate. In fact, the positions of the catalytic residues in the structures obtained for glutaminase dimers and tetramers are virtually identical[25,38]. Therefore, these findings further raised the possibility that GAC needs to form a higher-order oligomeric state in cells, resulting in a filament-like structure, in order for catalysis to occur[29,31,39]. Given the structural similarities between GAC and GLS2, it might be expected that GLS2 would undergo a similar structural transition, although GLS2 can in some cancers exhibit the opposite effects from GAC by acting as a tumor suppressor rather than a promoter raising the possibility that it might exhibit distinct structural features. Thus, we set out to see if a filament-like structure is indeed formed by GLS2 as well as GAC and whether its formation is coupled to catalysis.

Using negative stain electron microscopy (EM), we visualized the structural changes that recombinant wild-type human GAC and GLS2 undergo upon the addition of the substrate glutamine and the anionic activator Pi under catalytic conditions (Fig. 2a, b). GAC immediately formed filaments with the addition of the substrate and activator, similar in appearance to those described in a previous report[31], and the same was true for GLS2, thus indicating that filament formation is common to all glutaminase enzymes. However, once the substrate glutamine was fully converted to product (glutamate), the filaments disassembled into tetramers as shown for GLS2 in Fig. 2c, supporting the idea that filament formation is coupled to catalytic turnover.

To further establish a direct connection between filament formation and enzyme catalysis, we took advantage of right-angle light scattering (RALS) to monitor the assembly of these higher-order oligomers in real-time. Figure 2d shows an example for GLS2. Upon adding Pi and glutamine to the enzyme, there was an immediate, marked increase in light scattering. The increase in RALS peaked within minutes and was followed by a rapid decline, with the decrease in the peak matching the time scale for the activity assay and the dissociation of GLS2 filaments as visualized by negative stain EM. The maximal increase in RALS required both the substrate glutamine and the anionic activator Pi and was not detected in the absence of substrate (Supplementary Fig. 2a), demonstrating that the changes in RALS accurately reflect the association and dissociation of filaments. Subsequent additions of glutamine to the enzyme re-established filament formation, with substrate consumption again resulting in filament disassembly (Fig. 2d). The addition of the product, glutamate, was unable to trigger filament formation. Taken together, these results further indicate that filament assembly-disassembly is directly coupled to substrate binding and catalytic turnover.

Mutations that result in the constitutive activation of the glutaminase enzymes were then examined by negative stain EM and RALS for their effects on filament formation. Changing a key lysine residue (i.e., Lys320 in GAC and Lys253 in GLS2) to alanine within the activation loop gives rise to constitutively active enzymes, with their activity matching that of the wild-type enzyme measured in the presence of the anionic activator Pi[38]. The RALS profile and negative stain images showed that glutamine addition alone was sufficient to drive constitutively active forms of human GAC (K320A) and GLS2 (K253A) to form filaments, as the activation loop substitution effectively substitutes for the requirement of an anionic activator (Supplementary Fig. 2b–d). Again, the product glutamate was unable to drive filament assembly, nor did the presence of high concentrations of glutamate prior to glutamine addition have any effect on the ability of substrate to induce filament formation (Supplementary Fig. 2d). The lifetime of the higher-order oligomeric filaments, as readout by RALS, was a direct function of the substrate (glutamine) concentration, as shown for GLS2 (Supplementary Fig. 2e). Likewise, the length of the GLS2 K253A filaments that formed were a function of substrate concentration and consisted of as many as 25 tetramers (Supplementary Fig. 2f), most likely because the greater the substrate concentration, the longer the time required for its total consumption and thus the ability to increase the growth of the filaments. This is again consistent with filament assembly-disassembly being directly coupled to enzyme activation and catalytic turnover. We then reasoned that conditions that allow substrate binding and filaments to form, but do not permit product (glutamate) formation, should result in the persistence of these higher-order oligomers. Therefore, we tested this prediction by taking advantage of our earlier finding that substituting an essential tyrosine residue within the enzyme active site of GAC does not disrupt substrate (glutamine) binding but prevents its hydrolysis to glutamate[25,34]. As shown in Fig. 2e, the RALS signal for GAC (Y466W) did not diminish over time.

## Structural determinations of GAC and GLS2 filaments

We determined a 3.3 Å cryo-EM structure of the human GAC (Y466W) mutant which because it is defective for hydrolyzing glutamine to glutamate, assembled into a stable filament upon binding substrate and the anionic activator phosphate (Pi)[25,34]. Each successive tetramer is rotated ~51 degrees relative to the filament axis with the helical rise of ~68 Å (Fig. 3a, b). We also obtained a 3.3 Å cryo-EM structure of the constitutively active recombinant human GLS2 (K253A) mutant that formed a filament immediately after the addition of glutamine alone (and prior to substrate depletion), as this mutant does not require an anionic activator to be catalytically competent (Fig. 3c, d). The tetramers are arranged in a side-by-side architecture and each successive

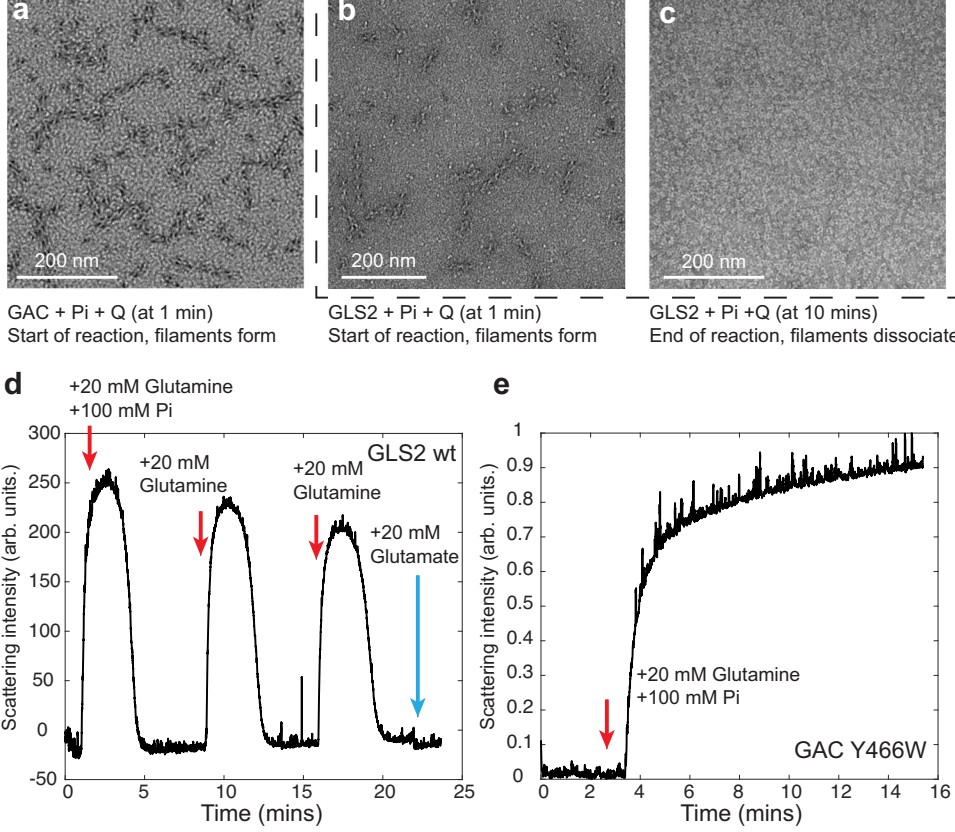

**Fig. 2 | The formation of the glutaminase filaments is directly coupled to catalytic activity. a** Negative stain EM ($n = 3$ independent images) of GAC (1 μM) with inorganic phosphate (Pi, 50 mM) one minute after adding glutamine (Q, 20 mM). **b** Negative stain EM ($n = 3$ independent images) of GLS2 (1 μM) with Pi (100 mM) one minute after adding glutamine (20 mM). **c** Negative stain EM ($n = 3$ independent images) of GLS2 (1 μM) with Pi (100 mM) 10 min after adding glutamine (20 mM). **d** Right-angle light scattering (RALS) of GLS2 (2 μM) with Pi (100 mM) after subsequent additions of glutamine (20 mM), or glutamate (20 mM). X-axis: time in minutes (mins); Y-axis: the intensity of the absorbance at 340 nm in arbitrary units (arb. units). **e** RALS ($n = 3$ independent experiments) of GAC (Y466W) with Pi (50 mM) and glutamine (20 mM). X-axis: time in minutes (mins); Y-axis: the intensity of the absorbance at 340 nm in arbitrary units (arb. units). Source data are provided as a Source Data file.

tetramer is rotated ~48 degrees relative to the filament axis, with the helical rise of ~66 Å (Fig. 3d). These structures for activated glutaminase enzymes differed from the previously proposed end-to-end double helix model for GAC filament formation[31]. In both the GAC and GLS2 filament structures, each tetramer interacts with its immediate neighbor through an interface of six alpha helices on the lateral side of the catalytic domain near the lid loop that closes over the substrate binding site (Fig. 3e, f). Within this highly conserved interface burying about 710 Å² and 752 Å² of solvent-accessible area on GAC and GLS2, respectively, Phe355 in GAC (Phe288 in GLS2), together with Phe373 (Phe306 in GLS2), Phe378 (Phe311 in GLS2), Asp412 (Asp345 in GLS2), and Gln416 (Gln349 in GLS2) from one subunit form an intricate interacting network with Asn375 (Asn308 in GLS2) and Gln379 (Gln312 in GLS2) from the subunit of an adjacent tetramer (Fig. 3g, h).

When comparing the filament structures with those for the corresponding tetrameric apo-enzymes described above (Fig. 1, and Supplementary Fig. 1), we found striking differences in the conformation of the activation loops and the lids. The activation loops in both the GAC and GLS2 tetramers, either in their apo-, substrate-bound, or product-bound states, are highly flexible and cannot be resolved, except when GAC is bound to the BTPES class of inhibitors (Supplementary Fig. 1). This led us to originally suspect that a flexible rather than stabilized activation loop was essential for the catalytic activity of these enzymes. However, in the substrate- and Pi-bound GAC filament, as well as in the constitutively active substrate-bound GLS2 filament structure, the activation loops are well resolved and adopt a conformation distinct from the inhibitor-induced state in GAC (Fig. 3e, f, and Supplementary Fig. 1).

The first significant conformational change involving the activation loops of the GAC and GLS2 filaments, compared to their positions in the apo-enzymes, resulted in a motif designated here as the 'phenylalanine-tyrosine lock' in GAC and the 'tyrosine-tyrosine lock' in GLS2. These residues lock in the substrate at the conserved active sites (Fig. 4a, b). In each dimer within a tetrameric unit comprising the filament structures of GAC and GLS2, Glu325 in GAC (Glu258 in GLS2) within the activation loop of one monomer subunit forms a salt bridge with Arg317 (Arg250 in GLS2) from the activation loop of the adjacent monomer subunit (Fig. 4c, d). In solving the cryo-EM structure for the GAC(Y466W) filament in the presence of glutamine and Pi, we were able to identify the Pi binding site within the dimer-dimer interface of each tetrameric unit where it interacts with Lys320 and thereby stabilizes the activation loop (Fig. 4c, and Supplementary Fig. 3a). This helps to explain how the binding of Pi to a mouse GAC(F327W) mutant (GAC F322W in human) is able to induce an increase in Trp327 fluorescence within the activation loops of the enzyme[40]. Conversely, in the absence of Pi and substrate, the activation loops of the enzyme are flexible and therefore are not visualized. Thus, the activation loops of each dimer within the filament fold in a compact manner, thereby bringing Phe318 of GAC (Tyr251 of GLS2) from each loop closer to the active site (Fig. 4c). In the apo-enzyme structures, Arg387 of GAC and Arg320 in GLS2 face toward the active site and prevent glutamine from being 'locked in'. However, in the filament structure of GAC, Pi

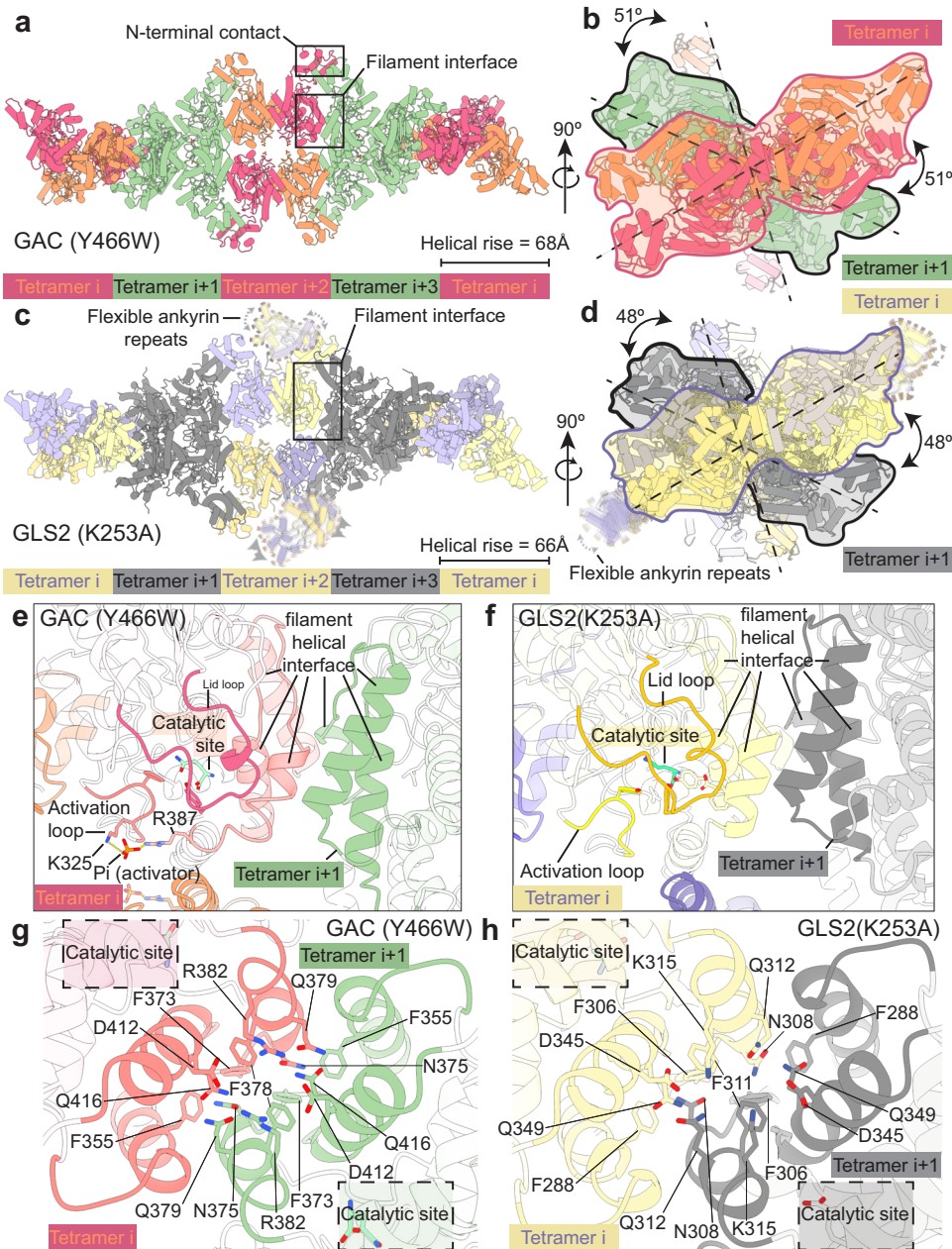

**Fig. 3 | Structures of the human GAC (Y466W) filament and the human GLS2 (K253A) filament captured under catalytic turnover conditions. a** Structure of the GAC (Y466W) filament showing a five tetramer stretch of the higher-order oligomer with a helical rise of 68 Å. Tetramers are colored alternately either orange and red or green. The orange and red tetramers are colored to distinguish between monomers in each tetramer. The filament interface and the N-terminal contacts are highlighted. **b** The GAC (Y466W) filament viewed down the filament axis. The first two tetramers are outlined with red and black silhouettes, sequentially. The dashed line indicates the longest dimension of each tetramer and the 51° rotation around the filament axis between tetramers. **c** Structure of the GLS2 (K253A) filament showing a five tetramer stretch of the higher-order oligomer with a helical rise of 66 Å. Tetramers are colored alternately either purple and yellow or gray. The purple and yellow tetramers are colored to distinguish between monomers in each

tetramer. Ankyrin repeats are not built due to their high flexibility and outlined with dashed profile. The filament interface is highlighted. **d** The GLS2 (K253A) filament viewed down the filament axis. The first two tetramers are outlined with purple and black silhouettes, sequentially. The dashed line indicates the longest dimension of each tetramer and the 48° rotation around the filament axis between tetramers. **e, f** The position of the helical filament interface, the activation loop, the catalytic site, and the lid loop in GAC (**e**, red) and GLS2 (**f**, yellow). The next tetramer in the filament is colored green in GAC and gray in GLS2. The helical filament interface is shown in solid colors. **g, h** The interaction network at the helical filament interface between tetramers for GAC (**g**, red, and green) and GLS2 (**h**, yellow, and gray). The close proximity of the catalytic site to the helical filament interface is highlighted with a dashed box.

interacts with Arg387, enabling it to form a salt bridge with Glu397 at the dimer-dimer interface (Fig. 4c, d), which together with the interactions of Pi with Tyr394 and Lys398 (Supplementary Fig. 4a), may be necessary for the glutaminase enzymes to undergo the dimer-to-tetramer transition, a prerequisite for filament formation and catalytic activity. Interestingly, in the filament structure for the constitutively

active GLS2 mutant, Arg320 (which corresponds to Arg387 in GAC) forms the same type of salt bridge in the absence of Pi, as that seen when Pi binds to GAC(Y466W). The formation of these Arg-Glu salt bridges, together with the substitution of a less bulky alanine residue for lysine, allows the movement of the activation loops into a space that otherwise would only occur when Pi binds, and likely explains why

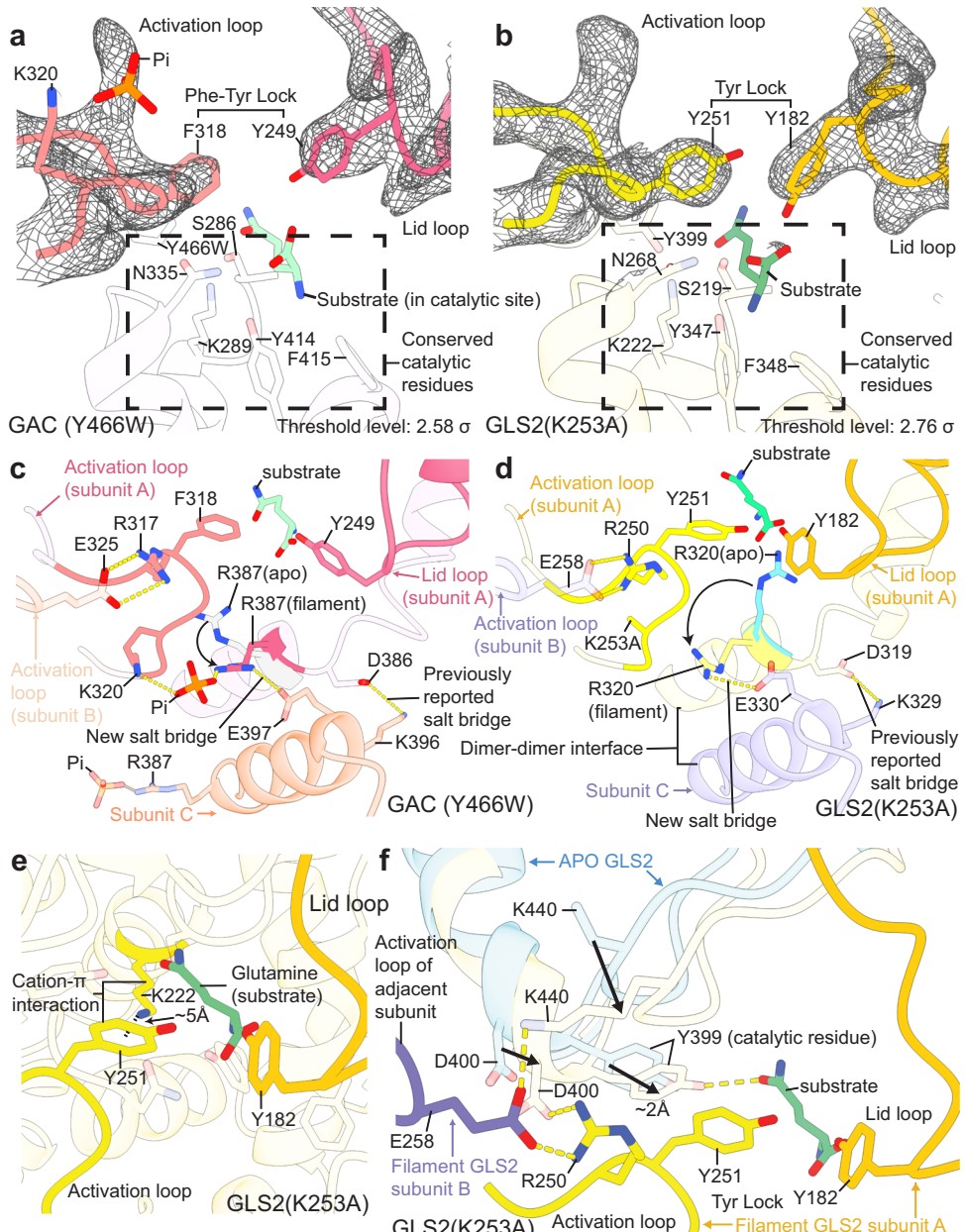

**Fig. 4 | Structures of the GAC and GLS2 filaments reveal allosteric conformational changes within the glutaminase tetrameric units during catalysis.**
**a**, **b** The phenylalanine-tyrosine lock in GAC (Phe318 and Tyr249) and the tyrosine-tyrosine lock in GLS2 (Tyr251 and Tyr182). Conserved catalytic residues are labeled, and the electron densities of the activation loop and the lid loop are shown as mesh. **c** The movement of Arg387 between apo-GAC (PDB ID: 3UNW, Arg387 is shown in light gray) and the GAC (Y466W) filament (pink) is shown with an arrow. The interactions between inorganic phosphate (Pi) and residues Lys320 and Arg387 near the activation loop in the GAC (Y466W) filament are highlighted. Salt bridges between GAC dimers within a tetramer (pink and salmon) are labeled as dashed yellow lines. **d** The movement of Arg320 between apo-GLS2 (Arg320 is shown in

cyan) and the GLS2 (K253A) filament (yellow) is shown with an arrow. Salt bridges between GLS2 dimers within a tetramer (yellow and purple) are labeled as dashed yellow lines. **e** The tyrosine-tyrosine lock in GLS2 consists of Tyr251 from the activation loop (yellow) and Tyr182 from the lid loop (orange). Glutamine is shown in green and the cation-π interaction between Lys222 and Tyr251 is labeled. **f** The movement of conserved catalytic residue Tyr399 between apo-GLS2 (light blue) and the GLS2 filament (light yellow) is shown with an arrow. The activation loops of two adjacent monomers within the GLS2 filament are colored yellow (subunit A) and purple (subunit B). Interactions between key residues consisting of hydrogen bonds and salt bridges are shown as yellow dashed lines.

the Lys-to-Ala substitutions within the loops yield enzymes that no longer require the binding of an anionic activator.

When comparing the filament structures with those of the apo-enzyme, a second significant conformational change involves the lid loops, resulting in the insertion of Tyr249 of GAC (Tyr182 of GLS2) into the active site pockets. These two conformational rearrangements involving the activation and lid loops fully close off the active site in each subunit, acting as a 'Phe-Tyr lock' in GAC and a 'Tyr-Tyr lock' in

GLS2, effectively locking the substrate glutamine into an optimal orientation proximal to the residues essential for catalysis (Fig. 4a, b). Moreover, the insertion of the phenyl ring of Phe318 of GAC (Tyr251 in GLS2) brings it within ~5 Å of Lys289 (Lys222 in GLS2), as shown for GAC in the Supplementary Fig. 3b and GLS2 in Fig. 4e, thereby enabling a cation-π interaction that orients and increases the basicity of a key active site lysine residue. This enables the lysine residue to effectively deprotonate an active site serine (i.e., Ser286 in GAC and Ser219 in

GLS2), which undergoes a nucleophilic attack and forms a tetrahedral intermediate with glutamine. Hydrolysis can then occur following this initial reaction to form glutamate and ammonia[41].

An additional conformational change that occurs specifically in the constitutively active GLS2 filament structure, when compared to the tetrameric apo-enzyme, results in a hydrogen bonding network introduced by the activation loops. Hydrogen bonds that form between Arg250 and Asp400 of GLS2 appear to 'stretch' the alpha helix containing Asp400, causing Tyr399 on the N-terminal end of the alpha helix to insert further into the active site where the substrate binds (Fig. 4f). Following the insertion of Tyr399, Glu258 from the adjacent dimer forms a hydrogen bond with Lys440, which displaces the loop containing Lys440, and therefore makes space for the insertion of Tyr399. As a result, Tyr399 is now in close proximity to both Lys222 and the substrate, which is necessary for catalysis. Interestingly, the corresponding tyrosine residue in apo-GAC is already inserted into the active site (Supplementary Fig. 4b), i.e., identical to its position in the GAC and GLS2 filament structures. Because this catalytic residue is properly positioned in the active site even in the apo- or 'resting' state of GAC, it might contribute, together with the absence of ankyrin repeats (see below), to GAC having a significantly higher specific activity compared to GLS2. Collectively, these visualizations of the interactions within the active sites of the GAC and GLS2 filaments upon the binding of substrate now provide insights into the mechanisms by which the glutaminase enzymes catalyze glutaminolysis.

## A mechanism for glutaminase activation coupled to filament formation

The high-resolution glutaminase filament structures and RALS data presented here show how glutaminase filament formation is coupled to catalysis. In the apo-enzymes, the lid loops do not adopt a fixed conformation and are highly flexible. Such flexibility would likely introduce steric clashes at the tetramer-tetramer interfaces within the filament, thereby preventing filament formation (Fig. 5a, apo state). Upon binding to the enzyme, the substrate forms hydrogen bonds with Glu381 in GAC and Glu314 in GLS2, resulting in the movement of these residues toward the active site (Supplementary Fig. 5a, b). As shown for GLS2, this enables the alpha helix containing residues Asn308 to Thr317 to move inward, resulting in additional interactions between Thr310 and Gln185, and between Ser313 and Pro184 on the lid loop (Supplementary Fig. 5a). These two sets of interactions, and those which occur between Tyr249 and Phe318 in GAC, and between Tyr182 and Tyr251 in GLS2, with glutamine stabilize the lid loop in a conformation that together with the activation loop effectively locks-in the substrate. This is necessary for catalysis, as well as exposes the filament interface, therefore allowing additional tetramers to contribute to the growth of the higher-order oligomer (Fig. 5a, substrate-bound state). Reciprocally, it is the binding of tetramers to form the enzyme filament that restricts the flexibility of the lids and allows them to adopt a conformation that closes over the substrate binding sites, thus enabling catalysis to occur. Following a catalytic turnover, the product glutamate no longer interacts with the lid loop tyrosine and the active site glutamic acid residues and thus has a relatively low affinity for the enzyme. Upon its dissociation, the activation loops and the lids become highly flexible again to await the next substrate binding event. When all the available substrate is consumed, the lids are not able to be restrained which disrupts the filament interface and gives rise to the dissociation of the filament back to tetramers (Fig. 5a, product released state). The missing density of the lid loop of the terminal tetramer in the cryo-EM reconstruction for the constitutively active GLS2 (K253A) filament indicates its flexibility, representing a snapshot of this detachment (Supplementary Fig. 5c) that occurs following catalytic turnover and product release. However, in the case of the GAC (Y466W) mutant, which binds Pi and substrate but cannot undergo catalysis, the filament persists (Fig. 2e) and now the density of

the lid loop within the terminal tetramer of the cryo-EM reconstruction remains intact (Supplementary Fig. 5d). We interpret the different occupancy of the lid loop density as the rigidity of helical symmetry between GAC and GLS2. In the more flexible GLS2 filament, the alignment error is greater at the distal regions of the cryo-EM map, resulting in lower local resolution for the terminal tetramers, thus causing weaker electron density occupancy of the lid loop in the reconstruction, while the local resolution of the GAC Y466W filament is more consistent across the whole map (Supplementary Figs. 9, 10).

Given that filament formation is directly coupled to glutaminase activation, it might then be expected that small molecule inhibitors that prevent the glutaminases from undergoing the conformational transitions necessary for their activation would negatively impact their ability to assemble into these higher-order oligomeric structures. We therefore examined the effects of two allosteric glutaminase inhibitors, UPGL0004 and 968, on filament formation. UPGL004 is a potent member of the BPTES-class of compounds and specifically targets GAC[27,36] by binding to the activation loops (Supplementary Fig. 1) and trapping them in a stable conformational state that prevents their movements into the active sites and from working together with the lid loops to establish the filament interface and lock in the substrate to undergo catalytic turnover. Thus, we found that while the vehicle control (DMSO) did not interfere with the ability of GAC (Y466W) to form filaments within 30 s which persisted through 30 min as visualized by negative stain EM (Supplementary Fig. 6a, and Fig. 5b), UPGL004 addition immediately blocked their assembly (Fig. 5c). The BPTES-class of compounds are ineffective against GLS2 because of subtle differences in their activation loops from those of GAC[24]. However, 968 is a small molecule inhibitor that effectively inhibits GLS2 catalytic activity[9]. It appears to bind at the activation loops of GLS2 based on the density we observed for the cryo-EM structure obtained for the 968-GLS2 complex, but without overlapping the binding sites for either the BPTES-class of inhibitors nor Pi (Supplementary Fig. 6b), consistent with kinetic analyses from our laboratory[14]. Rather, upon binding to GLS2, 968 would block the salt bridge that forms between Arg258 on one activation loop and Glu250 on an adjacent activation loop, thereby preventing the substrate from being locked in at the catalytic site. This could also potentially weaken the interactions between GLS2 dimers to form tetramers, as we have observed by cryo-EM 2D classifications where 968 treatment results in the appearance of some dimeric enzyme particles (Supplementary Fig. 6c). Collectively, these effects would be expected to prevent the formation of filaments that are required for optimal catalysis, which is what we observed by negative stain EM in Fig. 5d, e. Likewise, mutation of a key residue within the tetramer-tetramer interface necessary for filament formation (e.g., Asn308 in GLS2), significantly reduces their size to exist as tetramers (Fig. 5f, and Supplementary Fig. 6a) with a corresponding decrease in catalytic activity (Fig. 5g). Together, these findings further reinforce the conclusion that the formation of glutaminase filaments is an essential step for enzyme activation and to achieve maximal catalytic turnover.

## The role of the ankyrin repeat motif in GLS2

The C-terminal region of GLS2 contains an ankyrin repeat motif consisting of three pairs of helical bundles located on the distal ends of the tetramer, which is absent in GAC (Fig. 6a, b). The ankyrin repeats in GLS2 appear to exert a negative regulation of catalytic activity, despite being approximately 70 Å away from the active sites in each subunit. The KGA isoform of GLS also contains ankyrin repeats and has a much lower specific activity than its C-terminal truncated splice variant GAC; moreover, truncated forms of KGA lacking these repeats show increased specific activity[33]. Each GLS2 dimer within a tetrameric unit shares an ankyrin repeat bundle connected to the catalytic domain through a flexible linker. Interestingly, the ankyrin repeats asymmetrically occupy the area between the N-terminus of adjacent

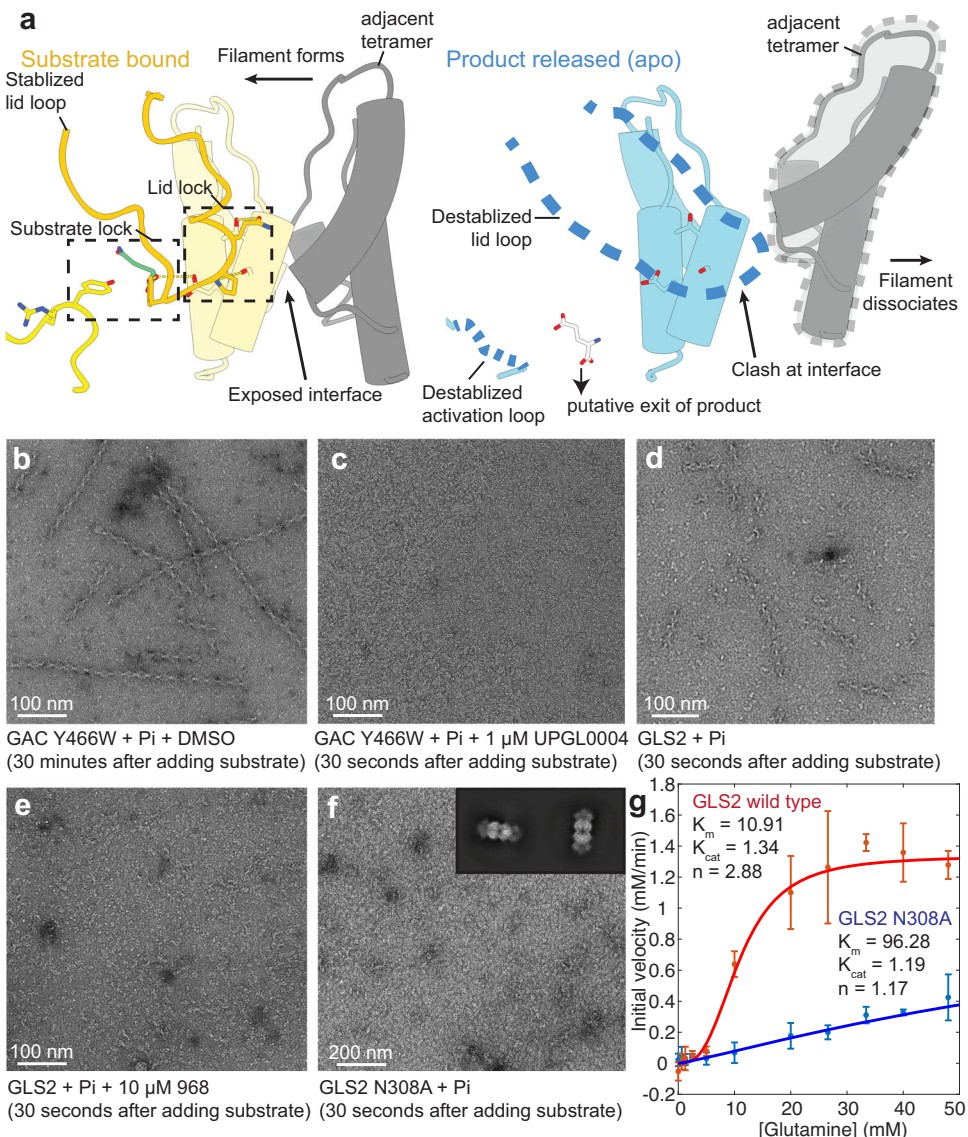

**Fig. 5 | The proposed mechanism of glutaminase filament formation and dissociation. a** The proposed mechanism for glutaminase filament formation and dissociation. Left: When substrate binds, the lid (yellow) forms a 'lid lock', and the helical interface is exposed (shown in gray), allowing the adjacent tetramer to extend the filament. Right: When product is released, the lid loop becomes flexible, displacing the adjacent tetramer, and blocking the helical interface. **b** Negative stain EM (*n* = 3 independent images) of GAC (Y466W) (1 μM) with Pi (200 mM) and DMSO 30 min after adding substrate (20 mM). **c** Negative stain EM (*n* = 3 independent images) of GAC (Y466W) (1 μM) with Pi (200 mM) and UPGL0004 (1 μM) 30 s after adding substrate (20 mM). **d** Negative stain EM (*n* = 3 independent images) of GLS2 (1 μM) with Pi (50 mM) 30 s after adding substrate (20 mM). **e** Negative

stain EM (*n* = 3 independent images) of GLS2 (1 μM) with Pi (50 mM) and compound 968 (10 μM) 30 s after adding substrate (20 mM). **f** Negative stain EM (*n* = 3 independent images) of GLS2 (N308A) (1 μM) with Pi (50 mM) 30 s after adding substrate (20 mM). Inset: 2D classes of cryo-EM indicates GLS2 (N308A) exists as tetramer with substrate (20 mM) and Pi (50 mM). **g** Activity assay (*n* = 3 independent experiments) of wild-type GLS2 (red) and GLS2 (N308A) (blue). X-axis: the concentration of substrate (mM); Y-axis: the initial velocity of enzyme (mM/min); $K_m$: Michaelis constant (mM); $K_{cat}$: maximum velocity (mM/min); n: Hill coefficient. Circles indicate the mean value and error bars are shown for the standard error. The curve is fit to the Michaelis–Menten equation. Source data are provided as a Source Data file.

monomeric units within a dimer, showing a preferential association for one monomer over the other. In addition, the ankyrin repeats for each dimer within a tetramer are predominantly positioned on the same side of the tetramer (Supplementary Fig. 7). The previously solved crystal structure of kidney-type glutaminase (KGA/GLS) shows a similar asymmetric positioning of the ankyrin repeats[33] (Fig. 1a). As described above, the activation loops within each dimer unit comprising the filament structures need to form an Arg-Glu salt bridge that enables the activation loops to be correctly positioned for catalytic activity (i.e., Arg317-Glu325 in GAC and Arg250-Glu258 in GLS2). This requires a scissor-like movement between two monomers within the same dimer to position their activation loops into close proximity

(Fig. 6a, b, insets at the bottom; these movements are indicated by arrows; Supplementary movie 1). However, within the GLS2 structure, the ankyrin repeats appear to hinder this movement and thereby may prevent the activation loops from adopting the positions necessary for maximal catalytic capability (Fig. 6b). These effects may help contribute to GLS2 having a lower specific activity compared to GAC.

## Discussion
The glutaminase enzymes GAC and GLS2 have both been implicated in cancer progression due to the upregulation of their expression and catalytic activities[9,10,14,19,20,42]. The elevations in glutamine metabolism triggered by these metabolic enzymes serve to compensate for the

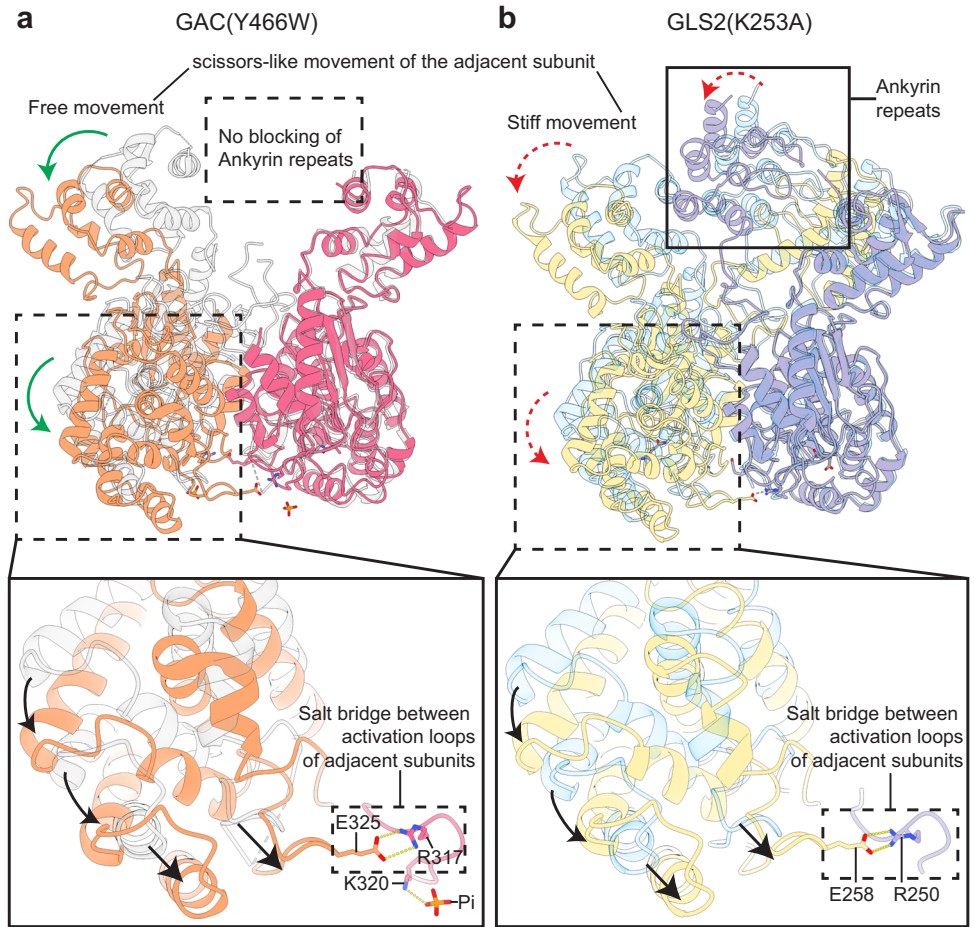

**Fig. 6 | The proposed mechanism for regulation of GLS2 catalytic activity by the C-terminal ankyrin repeats. a** The unrestricted scissor-like movement of the N-terminal domain between monomers within a dimer of the GAC filament (orange and magenta) and apo-GAC (light gray) after filament formation. The arrows indicate the direction of conformational changes. Inset on the bottom: the scissor-like movement brings Glu325 and Arg317 from adjacent GAC monomers within a dimer into close proximity and helps them form a salt bridge to stabilize the activation loops. **b** The scissor-like movement of the N-terminal domain between monomers within a dimer of the GLS2 filament (purple and yellow) and apo-GLS2 (light blue) is restricted by the ankyrin repeats (solid box). The arrows indicate the direction of conformational changes. The formation of the salt bridge between Glu258 and Arg250 from adjacent GLS2 monomers within a dimer is burdened by the blocking of ankyrin repeats, which prevents the activation loops contact at close proximity. Also, see Supplementary movie 1.

uncoupling of glycolysis from the TCA cycle, known as the Warburg effect, by providing the carbon sources necessary for the biosynthetic processes that enable cancer cells to undergo rapid proliferation and survive various types of cellular stress that would otherwise kill normal healthy cells. However, in some cancer contexts, GLS2 has been proposed to function as a tumor suppressor[17,18]. While this raised the question of whether GLS2 might exhibit distinct structural features from GAC, here we show that like GAC, GLS2 assumes a higher-order filament-like structure which is directly coupled to catalytic turnover.

In our efforts to learn more about the mechanisms by which the glutaminase enzymes catalyze glutamine hydrolysis, we had developed fluorescence spectroscopic readouts to monitor directly the binding of the substrate glutamine in the presence and absence of the anionic activator Pi, as well as probe the interactions of allosteric inhibitors while simultaneously assaying enzyme activity[27,37,38,40]. These studies showed how the binding of substrate and anionic activators helped to drive the transition between enzyme dimers to tetramers[25]. Given that the addition of an anionic activator-like Pi could induce the formation of glutaminase tetramers and was necessary for enzymatic activity, it was assumed that the dimer-to-tetramer transition was responsible for catalysis. Still, it was puzzling that when comparing the active sites of the dimeric and tetrameric GAC species, there were no apparent differences in the positioning of the key catalytic residues,

nor was there any indication of how the activation loops contributed to enzyme activity. However, we have been able to obtain important insights into the mechanisms of glutaminase activation and catalysis when examining the ability of GAC and GLS2 to form filament-like structures under catalytic conditions, both through the use of cryo-EM and RALS. By performing cryo-EM structural analyses of the GAC (Y466W) mutant which is still able to bind glutamine and Pi but is defective in hydrolysis, we were able to capture a snapshot of the substrate- and activator-induced conformational changes that lead to filament assembly. We found by negative stain EM, cryo-EM, and RALS that the constitutively active GLS2 (K253A) mutant also formed higher-order filament-like structures comprising several enzyme tetramers under catalytic turnover conditions, similar to GAC. Importantly, we demonstrated that the lifetimes of the filament structures for both GAC and GLS2 were directly coupled to catalysis, as they persisted as long as sufficient substrate was available but then rapidly dissociated when the substrate was ultimately used up, confirming suggestions that the formation of glutaminase filaments is directly linked to their catalytic activity[29,31,39].

Previously, it has been reported that various metabolic enzymes assemble into filamentous structures[28,43]. Several recent structural studies leveraging the advances in cryo-EM have demonstrated that filament assembly allows for allosteric regulation of enzymes[44-51]. Our

structures of glutaminase filaments demonstrate that filament formation stabilizes the allosteric sites on glutaminase (i.e., the activation loop) in a manner similar to that observed in phosphoribosyl pyrophosphate synthetase 1 (PRPS1)[46]. In the case of PRPS1, filament formation enabled binding of the activator, Pi. In contrast, we find that Pi, or the introduction of mutations that result in a constitutively active enzyme, is required for filament formation. Given that the presence of substrate is also required for glutaminase filament formation, we propose that filament formation stabilizes the active form of glutaminase, similar to what has been suggested for the enzyme CTP synthase 1[45].

For all prior structures of the glutaminase isozymes, the activation loops were only visible when the enzymes were bound to allosteric inhibitors of the BPTES family of compounds and there were no indications of how they play a critical role in enzyme catalysis, especially given their apparent distance from the active sites. There also was little indication of how the activation loops communicated with the lids that form over the bound substrate, which is necessary for enzyme catalysis[34,38]. However, the cryo-EM structures of activated GAC and GLS2 filaments show activation loops that are well resolved and move toward the active site of each enzyme subunit, positioning an essential tyrosine residue sufficiently close to a catalytically important lysine residue to form a cation-π interaction. This leads to the de-protonation of an active site serine residue, which is then poised for nucleophilic attack on the substrate glutamine. The glutaminase filament structures also show that the lids undergo conformational changes relative to their positions in the apo-enzyme (tetramer), resulting in the insertion of a key tyrosine or phenylalanine residue into the active site. These changes effectively close off each active site, locking the substrate glutamine into the proper orientation for catalysis. Several additional hydrogen bonding interactions are evident within the filament structures that help stabilize the interactions between the tetrameric units and enable the activation loops to be inserted into the active sites. Upon catalytic turnover, the release of the product glutamate relieves constraints on the activation loops and introduces flexibility to the lid loops, which disrupts the interface between tetramers and results in the dissociation of the filaments, thus accounting for the tight correlation between enzyme activity and filament formation/dissociation.

Our structure-function studies of glutaminase filaments have shed light on how anionic activators like Pi bind to activation loops of the enzymes and drive a dimer-to-tetramer transition and filament assembly which are essential steps for catalysis. They also highlight how allosteric inhibitors targeting the glutaminase enzymes and the presence of ankyrin repeats prevent the activation loops from undergoing the conformational changes necessary for catalytic activity and filament formation. Moreover, these findings now open the way to several interesting new lines of study. For example, new therapeutic strategies might be developed to block the formation of these higher-order oligomers. Identifying other potential activators of both GAC and GLS2 in cancer cells that can induce filament formation will be important, given that the Pi levels necessary to stimulate enzyme activity and filament formation might not be achievable under many physiological contexts. It will also be interesting to see how post-translational modifications such as lysine succinylation, which occur at specific sites on GAC and have been suggested to both promote enzyme activation by stimulating oligomer (tetramer) formation[29] and to enhance ubiquitination and degradation[52], affect filament formation, as well as whether the ankyrin repeats are responsible for conferring GLS2 with functional capabilities distinct from those of GAC in different cancers. Finally, an intriguing possibility will be to see whether the glutaminase enzymes co-assemble in filamentous structures with other enzymes suggested to form higher-order oligomers such as glutamate dehydrogenase[28]. Such assemblies might more efficiently couple the activities of members of a metabolic pathway by substrate channeling, with the products of one enzyme being channeled to the next enzyme in the pathway to use as substrates.

## Methods

### Recombinant glutaminase expression and purification
An N-terminal His-tagged form of the full-length human liver-type glutaminase isoform GLS2 without the mitochondrial localization sequence (residue 38-602) was cloned into the pET28a plasmid[36]. Site-directed mutagenesis was performed using Phusion DNA polymerase (New England Biolabs, NEB). The GLS2 primers (5′–3′) used to mutate residue N308A were TACATGGGGTTTCAGCGCGGCCACA (Forward) and GAATGTGGCCGCGCTGAAACCCATGTA (Reverse). Protein expression and purification of GLS2 was carried out first by transforming constructs into *E. Coli* BL21 (DE3) competent cells (NEB), which were then grown in LB media overnight with 50 µg/mL kanamycin. The starter cultures were subsequently inoculated at -1:100 ratio in 6 L cultures with the same antibiotic concentration and left shaking at 37 °C, 180 rpm for 3–4 h until the OD600 reached between 0.6 and 0.8. The flasks were then chilled at 4 °C for 1–2 h before induction with 30 µM IPTG (isopropyl β-D-1-thiogalactopyranoside) and shaking at room temperature (RT), 180 rpm for 16–18 h. Cells were collected by centrifugation at $5000 \times g$ for 10 mintes and frozen before resuspension in 150 mL lysis buffer (50 mM Tris−HCl pH 8.5, 500 mM NaCl, 10% glycerol), supplemented with protease inhibitor cocktail (Roche). Cells were lysed with lysozyme and mechanical sonication, and DNase I was added to reduce the mixture's viscosity. The soluble fractions were separated from the debris by ultra-centrifugation (Ti45 rotor, $186,368 \times g$ for 45 min). The lysate was then loaded onto $Co^{2+}$-charged TALON resin (GoldBio), equilibrated previously with the wash buffer (50 mM Tris−HCl pH 8.5, 10 mM NaCl, 10 mM imidazole). The protein that bound to the column was washed with the wash buffer and eluted with wash buffer supplemented with 320 mM imidazole. Further purification was performed by anion exchange chromatography using HiTrap Q HP column (Cytiva) and size-exclusion chromatography using Superdex 200 pg 16/600 column (GE Healthcare). Human GAC was purified in the same manner as GLS2[25,27]. Proteins were kept in 20 mM Tris−HCl pH 8.5, 150 mM NaCl, snap-frozen in liquid nitrogen, and stored at −80 °C. Protein concentrations were determined by absorbance at 280 nm using extinction coefficients calculated using the Expasy ProtParam tool.

### Right-angle light scattering (RALS)
RALS method is adapted from a previously published paper[45]. Frozen glutaminase aliquots were thawed on ice and diluted to a final concentration of 2 µM in 1 mL of gel filtration buffer (20 mM Tris−HCl pH 8.5, 150 mM NaCl) added in a 1.2 mL cuvette and inserted into a Varian Cary Eclipse fluorimeter. The cuvette temperature was set to 25 °C and stirred with a magnetic stir bar. The signal was recorded using excitation and emission wavelengths of 340 nm (5 nm bandpass). The samples were incubated until the signal was stable (-2–3 mins), then 100 µL of glutamine from a 200 mM stock solution were added to initiate the reaction. Subsequently, 100 µL of inorganic phosphate (1 M) or glutamate (200 mM) were added for further analysis. The raw scattering traces were filtered to improve the signal-to-noise ratio using SciPy (version 1.9.2) and plotted using Matplotlib (version 3.7.1) and MATLAB (version R2022b).

### Glutaminase activity assays
Glutaminase activity was measured by coupling NAD+ reduction to glutamine hydrolysis in a two-step reaction, previously described by literature[25]. Each individual experimental group has its own blank control, which was prepared by adding 3 M HCl immediately before adding enzyme to the substrate and Pi to subtract any possible signal introduced by glutamine intrinsic hydrolysis in the solution. The curves of initial velocity of glutamine hydrolysis as a function of

substrate concentration were fit to the Michaelis–Menten equation $v = (K_{cat} S^n)/(K_m + S^n)$, where $v$ is the initial velocity (Y-axis), $S$ is the concentration of substrate (X-axis), $K_{cat}$ is the maximal velocity at saturating concentration of glutamine, $K_m$ is the Michaelis constant, and $n$ is the Hill coefficient. Triplicates were performed and the curves were fit and plotted using Matlab R2022b.

### Negative stain electron microscopy

Formvar/carbon film 200 mesh copper grids (Electron Microscopy Sciences, EMS) were plasma cleaned by using PELCO easiGlow system (TED PELLA). Glutaminase (final concentration: 1 µM) was mixed with the substrate glutamine (final concentration: 20 mM). Inorganic phosphate ($K_2HPO_4$ final concentration: 50 mM) was added to the mixture when needed. The mixture was incubated at RT for 30 s. Then, 10 µL of the mixture were applied onto the grids for a 60-s incubation, and the excess protein solution was blotted with filter paper. Ten µL of 2% uranyl acetate were then applied to the grid twice for two 30-s stainings followed by blotting. The grids were air-dried for 5 min and visualized by Thermo Fisher F200Ci electron microscope at 120 keV.

### Cryo-EM grids preparation and images acquisition for apo-GLS2

Four µL of 5 µM GLS2 were applied to plasma-cleaned grids (300 mesh UltraAu Foil R1.2/1.3, Electron Microscopy Sciences, EMS) for a 30-s incubation. The excess solution was blotted for 4–6 s with filter paper by a Vitrobot mark IV (Thermo Fisher Scientific). Grids were subsequently vitrified in liquid ethane and stored in liquid nitrogen. Cryo-EM single particle micrographs of apo-GLS2 were collected on a Talos Arctica (Thermo Fisher Scientific) at the Cornell Center for Materials Research. The microscope was operated at 200 keV at 63,000× nominal magnification using a Gatan K3 direct electron camera in super-resolution mode, with a Gatan GIF Quantum LS Imaging energy filter, corresponding to a physical pixel size of 1.23 Å/pix. Given the orientation bias of the sample, a non-tilted dataset of 1171 images and a 30-degree tilted dataset of 488 images were collected and merged for further data processing. The images were obtained with a defocus range of −0.8 to −2.0 µM by EPU (Thermo Fisher Scientific). Each stack movie was recorded with 50 frames for a total dose of ~50 e⁻/Å². 

### Cryo-EM grids preparation and images acquisition for GLS2 (K253A) filament

Human GLS2 (K253A), 3 µM, was mixed with 20 mM glutamine to initiate filament formation. The mixture was applied to plasma-cleaned grids (300 mesh Quantifoil Au R1.2/1.3, EMS) for 30-s incubations. The excess solution was blotted for 5–7 s with filter paper by a Vitrobot mark IV. Grids were subsequently vitrified in liquid ethane and stored in liquid nitrogen. Cryo-EM single particle micrographs of apo-GLS2 were collected on a Titan Krios (Thermo Fisher Scientific) at The New York Structural Biology Center. The microscope was operated at 300 keV at 81,000× nominal magnification using a Gatan K3 direct electron camera in counting mode, corresponding to a physical pixel size of 1.058 Å/pix. Thirty-eight hundred and four images were obtained with a defocus range of −0.8 to −2.0 mm by Leginon[53]. Each stack movie was recorded with 40 frames for a total dose of ~50 e⁻/Å².

### Cryo-EM grids preparation and images acquisition for GAC (Y466W) filament

Human GAC (Y466W), 8 µM, was mixed with 20 mM glutamine and 1 M inorganic phosphate to initiate filament formation. The mixture was applied to plasma-cleaned grids (300 mesh Quantifoil Cu R1.2/1.3, EMS) for 30-s incubations. The excess solution was blotted for 2–3 s with filter paper by a Vitrobot mark IV. Grids were subsequently vitrified in liquid ethane and stored in liquid nitrogen. Cryo-EM single particle micrographs of GAC (Y466W) filament were collected on a Talos Arctica (Thermo Fisher Scientific) at the Cornell Center for Materials Research. The microscope was operated at 200 keV at

63,000× nominal magnification using a Gatan K3 direct electron camera in counted mode, with a Gatan GIF Quantum LS Imaging energy filter, corresponding to a physical pixel size of 1.31 Å/pix. Thirty-nine hundred images were obtained with a defocus range of −0.8 to −2.0 mm by SerialEM[54]. Each stack movie was recorded with 50 frames for a total dose of ~50 e⁻/Å².

### Cryo-EM grids preparation and images acquisition for GLS2 dimer induced by the inhibitor 968

Four µL of 5 µM GLS2 were treated with 30 µM 968 inhibitor in 2% DMSO, applied to plasma-cleaned grids (300 mesh UltraAu Foil R1.2/1.3, Electron Microscopy Sciences, EMS) for a 30-s incubation. The excess solution was blotted for 4–6 s with filter paper by a Vitrobot mark IV (Thermo Fisher Scientific). Grids were subsequently vitrified in liquid ethane and stored in liquid nitrogen. Cryo-EM single particle micrographs of 968-GLS2 were collected on a Talos Arctica (Thermo Fisher Scientific) at the Cornell Center for Materials Research. The microscope was operated at 200 keV at 63,000× nominal magnification using a Gatan K3 direct electron camera in super-resolution mode, with a Gatan GIF Quantum LS Imaging energy filter, corresponding to a physical pixel size of 1.23 Å/pix. Given the orientation bias of the sample, a non-tilted dataset of 935 images and a 30-degree tilted dataset of 719 images were collected and merged for further data processing. The images were obtained with a defocus range of −0.8 to −2.0 µM by EPU (Thermo Fisher Scientific). Each stack movie of non-tilted dataset and 30-degree tilted dataset was recorded with 50 frames for a total dose of ~38 e⁻/Å² and ~48.5 e⁻/Å², respectively.

### Cryo-EM data processing

For the apo-GLS2 map, dose-fractionated image stacks were subjected to motioncor2 wrapped in RELION 4.0.0[55–57] through SBGrid, followed by patch CTF estimation in CryoSPARC V4.0.0[58]. The motion-corrected micrographs with CTF better than 4 Å were passed for later processing. A blob picker in CryoSPARC was used to generate the templates for further Topaz picking[59], and ~1 M particles were picked. The particles were subjected to 3D classification by using ab initio reconstruction and heterogenous refinement with four classes, and only the best classes were kept. The remaining ~329 K particles were subjected to homogenous and heterogenous refinement. After several rounds of refinement, CTF refinement was carried out for per-particles and per-exposure groups. The final reconstruction with C2 symmetry yields a map with a resolution of 3.12 Å determined by FSC. The quality of the map was improved by Phenix using density modification. The local resolution map is generated by CryoSPARC. Cryo-EM maps are visualized in ChimeraX[60]. The workflow is provided in Supplementary Fig. 8.

For the GAC (Y466W) filament structure, dose-fractionated image stacks were subjected to patch motion correction, followed by patch CTF estimation in CryoSPARC V4.2.1. The motion-corrected micrographs with CTF better than 4 Å were passed for later processing. The template-free filament trace job in CryoSPARC picks ~397 K particles for 2D classification. About 180 K particles from the best 2D classes were extracted and subjected to 3D classification by using ab initio reconstruction and heterogenous refinement with four classes. The best (~112 K) particles from 3D classification were picked and subjected to multiple rounds of homogenous and non-uniform refinement with CTF refinement for per-particles and per-exposure group, which yielded a 4 Å map. An atomic model of GAC tetramer was docked into the map to measure the helical parameters for further helical reconstruction. Then using helical reconstruction in CryoSPARC, the helical twist was searched between 40–60 degrees, and the helical rise was searched between 55–85 Å, which yielded a 3.6 Å map. Another round of 3D classification was performed to further select for the best particles. About 110 K particles were selected to undergo multiple rounds of helical refinement with D2 symmetry and CTF refinement and

yielded a 3.3 Å map determined by FSC. The optimized helical parameters estimated by CryoSPARC measures a 51° twist and a rise of 68 Å. Finally, the map was sharpened in CryoSPARC. The local resolution map is generated by CryoSPARC. The workflow is provided in Supplementary Fig. 9.

For the GLS2 (K253A) filament structure, dose-fractionated image stacks were subjected to patch motion correction, followed by patch CTF estimation in CryoSPARC V4.0.0. The motion-corrected micrographs with CTF better than 4 Å were passed for later processing, which yielded 3640 micrographs. Manual picking in CryoSPARC was used on 40 micrographs to pick ~600 particles manually, which yielded five classes of templates by 2D classification. The selected template was used for the filament trace job in CryoSPARC to template pick particles. 333 K particles were extracted and subjected to 2D classification. 92 K particles from the best 2D classes were extracted and subjected to 3D classification by using ab initio reconstruction and heterogenous refinement with four classes. The best class with ~48 K particles was subjected to multiple rounds of homogenous and non-uniform refinement with CTF refinement for per-particles and per-exposure group. Then using helical reconstruction with C1 symmetry in CryoSPARC, the helical twist was searched between 40–60 degrees, and the helical rise was searched between 55–85 Å, which yielded a 3.26 Å map determined by FSC. The optimized helical parameters estimated by CryoSPARC measures a 48° twist and a rise of 66 Å. Finally, the quality of the map was improved by Phenix using density modification and local sharpening. The local resolution map is generated by CryoSPARC. The workflow is provided in Supplementary Fig. 10.

For the 968-GLS2 dimer map, dose-fractionated image stacks of 0°-tilt and 30°-tilt datasets were separately subjected to patch motion correction, followed by patch CTF estimation in CryoSPARC V4.0.0. The motion-corrected micrographs with CTF better than 4 Å were passed for later processing, which yielded 709 micrographs for non-tilted dataset, and 895 for 30-degree tilted dataset. A blob picker in CryoSPARC was used to generate the templates for further Topaz picking, and ~1.7 M particles were picked for non-tilted dataset, ~1.2 M particles were picked for 30-degree tilted dataset. The particles were subjected to 2D classification and only the dimer classes were kept. The remaining ~132 K particles from non-tilted dataset and ~131 K particles from 30-degree tilted dataset were subjected to homogenous and heterogenous refinement. After several rounds of refinement, CTF refinement was carried out for per-particles and per-exposure groups. The final reconstruction with C1 symmetry yields a map with a resolution of 3.65 Å determined by FSC. The quality of the map was improved by Phenix using Auto-sharpening tool. The local resolution map is generated by CryoSPARC. Cryo-EM maps are visualized in ChimeraX. The workflow is provided in Supplementary Fig. 11.

### Model building and refinement

The initial apo-GLS2, GAC (Y466W), and GLS2 (K253A) models were adapted from the PDB entry 4BQM for the catalytic domain and 5U0K for the ankyrin repeats. First, all models were manually docked into the map by UCSF ChimeraX. Next, the model was fitted into the map by ISOLDE for the overall adjustment[61], then underwent multiple rounds of automatic refinement using Phenix real space refinement[62], and manual adjusting and building ligands using Coot[63]. Finally, validation was performed with MolProbity[64]. The final refinement statistics is provided in Supplementary Table 1. The model-to-map fittings and the quality of the maps are shown in Supplementary Fig. 12a for apo-GLS2, Supplementary Fig. 12b for GLS2 K253A filament, and Supplementary Fig. 13 for GAC Y466W filament, respectively. The model of 968-dimer was not built, and the quality of the maps of 968-GLS2 dimer is shown in Supplementary Fig. 6b.

### Reporting summary

Further information on research design is available in the Nature Portfolio Reporting Summary linked to this article.

## Data availability

The cryo-EM density maps and the atomic models for apo-GLS2, GAC (Y466W), and the GLS2 (K253A) filament structure reported in this paper have been deposited in EM Data Bank with codes EMD-40920 (Apo-GLS2), EMD-40918 (GAC Y466W) and EMD-40950 (GLS2 K253A), and to the Protein Data Bank with codes 8SZL (Apo-GLS2), 8SZJ (GAC Y466W), 8T0Z (GLS2 K253A). The cryo-EM density map for 968-GLS2 dimer is deposited in EM Data Bank with code EMD-43533. Source data are provided with this paper.

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

## Acknowledgements

We thank Drs. Katherine Spoth, Mariena Silvestry-Ramos, Kasahun Neselu, and Edward Eng for help with EM instrumentation. We would also like to thank Drs. Brian R. Crane and Sekar Ramachandran for the helpful feedback on the structural analysis, and Samuel L. Deck and Megan Xu for helping produce glutaminase mutants. This work was supported by grants from the NIH (R35GM152206, R01CA201402, and R01CA223534 to R.A.C.). The acquisition of the human apo-GLS2 and human GAC (Y466W) structures used the Cornell Center for Materials Research Shared Facilities, which is supported by NSF (DMR-1719875). The collection of negative stain images relied on using an instrument supported by the NIH through award S10OD030470-01. The acquisition of the human GLS2 (K253A) structure was performed at the National Center for Cryo-EM Access and Training (NCCAT) and the Simons Electron Microscopy Center located at the New York Structural Biology Center, supported by the NIH Common Fund Transformative High-Resolution Cryo-Electron Microscopy program (U24 GM129539) and by grants from the Simons Foundation (SF349247) and NY State Assembly.

## Author contributions

All authors contributed to each aspect of the study, but primarily, T-T.T.N., S.K.M, C.A. and S.F. developed methods to express and purify the glutaminase and mutants; C.A. and S.F. developed and performed the RALS assay; S.F. performed the activity assay; S.F. prepared the samples for negative stain EM and cryo-EM, and collected and processed the EM data. S.F. built the apo-GLS2, GAC Y466W filament, and GLS2 K253A filament structures; S.F., C.A., T-T.T.N., S.K.M. and R.A.C. analyzed the structures; S.F., C.A., T-T.T.N., S.K.M. and R.A.C. wrote the manuscript; R.A.C. conceived of and oversaw the project.

## Competing interests

The authors declare no competing interests.
