## [Peer Review File · Nature Communications]

Filament formation drives catalysis by glutaminase enzymes important in cancer progressionREVIEWER COMMENTS

Reviewer #1 (Remarks to the Author):

This manuscript reports structures of two different glutaminase isoforms that assemble into filamentous polymers, and provides convincing evidence that the filaments are assembled under active enzymatic conditions. The structures not only provide insight into the mechanisms of filament assembly, but also new structural views of the catalytic mechanism that should be of interest to the field and of significant impact given the role of glutaminase activity in promoting cancer cell survival. Overall the work is rigorous, clearly presented, and likely of broad interest.

I have one major concern about the causal link between polymerization and activity. While there is a clear correlation between enzyme activity and filament assembly, the evidence that assembly is required for activity is weaker and relies on the N308A assembly-defective mutation with a single time course activity measurement (Fig. 5g and Ext. Data 6c). I suggest that the authors:

- 1) Provide evidence that GLS2-N308A remains a tetramer (and doesn't dissociate into dimers) is important to know that disruption of filament assembly is the only effect of the mutation (the negative stain image in Fig. 5g doesn't seem sufficient – 2-D averages from cryo-EM images, or some orthogonal measure of quaternary structure could also show this).
- 2) Perform substrate kinetics of WT and the N308A mutant, to define what kinetic properties (K_M or k_{cat} ?) are affected by the loss of polymerization.

I have a few minor questions and suggestions about how the cryo-EM analysis was performed and is presented, as detailed below, but nothing that calls into question the major findings of the paper.

1) Ext. Data Fig. 5 - The density shown for the "terminal subunit" in the GAC and GLS2 filaments (right panels). It's not clear to me what "terminal subunit" means in this context. Is this just the last tetramer in the reconstruction? If so, then any differences seen between these subunits and subunits near the core would only be the result of heterogeneity in the dataset (likely due to flexibility of the filaments), as tetramers in that position would still be physically in the middle of a filament. If this is the case, then the rationale with regard to the relative ordering of the "terminal" subunit is not sound here, and should be omitted (but, the case for the conformational differences made in the right panel remains sound).

On the other hand, if efforts were made to identify physical ends of filaments from within micrographs and reconstruct those separately, this process should be described in results and methods and the map for this reconstruction should be described in the data table and deposited to EMDB.

2) Image processing and modeling with respect to the ankyrin repeat domains of GLS2. For both the free and filament forms of GLS2, the ankyrin repeat domains appear very poorly ordered in the cryo-EM maps (in the extended figures). It would be helpful to provide images of how the models fit into the maps in this region – even low pass filtered maps that provided some confidence that the overall position is correct would be useful.

One concern I have is with the symmetry applied in the reconstructions. Is there sufficient evidence that free GLS2 tetramers actually have C2 symmetry with the ankyrin repeats asymmetrically positioned within dimers? If the ankyrin repeats in the two halves of the tetramer were moving independently, then it should be possible to classify out different states. Similarly for the filament reconstructions, where now the ankyrin repeats appear to be on the same side in each tetramer, a more careful treatment of heterogeneity could provide greater confidence on the position of these domains.

3) The description of symmetry imposition in general is not very clear. At what steps, for example, were helical symmetry applied in the reconstructions as implied in the flowcharts for the filament reconstructions? Or were helical restraints imposed during refinement but helical symmetry not imposed? Why was point group symmetry not applied to the GLS2 filament reconstruction which, if I understand correctly, has a 2-fold symmetry axis coincident with the helical axis? For the GAC filament, why was C2 symmetry applied (and in which orientation)? If I understand correctly, the GAC filament should actually have D2 symmetry with 2-fold axes coincident with the helical axis and perpendicular to the axis. I think that these issues can generally be resolved with a more careful description of the approach taken and the rationale in the methods section.

I would also suggest, given the recent body of work describing mechanisms by which other enzymes are regulated by polymerization into filaments, that the authors include a brief discussion of how their results fit into this field, comparing how polymerization affects glutaminase activity with how it appears to affect activity and regulation in other enzymes. Of course, this is merely a suggestion that might help to broaden the appeal beyond the audience that will be interested in glutaminase regulation specifically.

Reviewer #2 (Remarks to the Author):

In this study, Feng et al. have taken an integrated, multidisciplinary approach incorporating structural, biochemical, and biophysical tools to unravel the molecular mechanisms of the GAC and GSL2 enzymes. To begin exploring the composition, architecture, and functionality of these proteins, the authors first demonstrated the ability of these glutaminase enzymes to form high-order assemblies and determined the cryoEM structures of GAC and GSL2 filaments at 3.35 and 3.26 angstrom resolutions, respectively.

Within these assemblies, the basic building block of the helical structure is comprised of a tetramer. This unit then interacts with neighboring tetramers, which results in helical stacking of GAC and GSL2 tetramers and elongation of filaments. The authors further show that these filaments contain novel structural features that are critical for polymerization. Similar to other metabolic and filamentous enzyme structures, GAC and GSL2 robustly polymerize into filaments to exert catalytic activity. In addition to modulating enzyme activity through polymerization, an unexpected feature of these filaments was the involvement of ankyrin domains in the regulation of enzyme activity. In summary, this study provides novel insight into the mechanism of polymerization of GAC and GSL2 enzymes and the regulation of their catalytic activity.

More detailed comments:

1. Between lines 55 and 57, the authors claim that the importance of higher-order structures of enzyme regulation and function has not been definitively established. However, there are many published studies (especially within the past few years) describing the regulation of enzyme activity by filament formation. The authors should introduce or discuss these studies in their manuscript and broaden their analysis of filamentous enzymes.
2. How do their insights into the enzyme assembly and regulation of catalytic activity by filament formation compare with other published studies?
3. To validate their structural models, the authors should characterize the residues located at the oligomerization interfaces by mutagenesis and functional studies and discuss the molecular mechanism of filament assembly.
4. More detailed description of cryoEM data processing strategy is needed in the methods section. Did the authors use local refinement, CTF refinement, or any other strategies to process their micrographs? If yes, the data processing scheme should be discussed in the methods section.
5. How were the helical parameters determined? Did the authors use cryoEM helical reconstruction tools? The details should be included in the methods section.
6. All three models have high clash scores for the reported resolution range. The authors should re-refine their structures and fix these errors. The rotamer and Ramachandran outliers should also be fixed and the validation reports should be updated.

Minor comment:

1. Typo: Thermo Fisher Scientific was misspelled throughout the methods section.

We sincerely thank the reviewers for their insightful comments and general support of our study. We have revised the manuscript to address their concerns. Our point-by-point response to each of the reviewers' comments is provided below.

Reviewer #1:

This manuscript reports structures of two different glutaminase isoforms that assemble into filamentous polymers, and provides convincing evidence that the filaments are assembled under active enzymatic conditions. The structures not only provide insight into the mechanisms of filament assembly, but also new structural views of the catalytic mechanism that should be of interest to the field and of significant impact given the role of glutaminase activity in promoting cancer cell survival. Overall the work is rigorous, clearly presented, and likely of broad interest.

I have one major concern about the causal link between polymerization and activity. While there is a clear correlation between enzyme activity and filament assembly, the evidence that assembly is required for activity is weaker and relies on the N308A assembly-defective mutation with a single time course activity measurement (Fig. 5g and Ext. Data 6c). I suggest that the authors: 1) Provide evidence that GLS2-N308A remains a tetramer (and doesn't dissociate into dimers) is important to know that disruption of filament assembly is the only effect of the mutation (the negative stain image in Fig. 5g doesn't seem sufficient – 2-D averages from cryo-EM images, or some orthogonal measure of quaternary structure could also show this).

As requested by this reviewer, we performed single-particle cryo-EM analysis on the GLS2-N308A mutant, and the results demonstrate that the GLS2-N308A mutant indeed remains a tetramer, as shown in Fig. 1a, below. For comparison, the difference between the 2D classes of GLS2 tetramers and dimers can be found in Extended Data Figure 6 in the revised manuscript. Importantly, no filament formation was observed in the micrographs (Fig. 1b below). We now include these data as the inset of Fig. 5f in the revised manuscript, and added the following text starting on line 311: “Likewise, mutation of a key residue within the tetramer-tetramer interface necessary for filament formation (e.g., Asn 308 in GLS2) prevents filaments from forming causing the enzyme to remain in the tetrameric state (Fig. 5f, and Extended Data Fig. 6a, right panel), with a corresponding decrease in catalytic activity (Fig. 5g in the manuscript).”

2) Perform substrate kinetics of WT and the N308A mutant, to define what kinetic properties (KM or Kcat?) are affected by the loss of polymerization.

To determine the kinetic parameters impacted by filament formation, we performed glutaminase activity assays comparing GLS2 WT and the GLS2 N308A mutant. As shown in Fig. 2, below, we found that the K_m value for the N308A mutant is significantly greater than that for the wild-type (WT) enzyme, while the K_{cat} values appear to be unaffected, as the overall trend suggests the activity of the N308A mutant reaches the same maximal activity as the WT enzyme at high substrate concentrations. This result supports our model that filament formation enhances substrate binding by promoting the formation of the Tyr-Tyr lock between the lid loop and the activation loop, but once the Tyr-Tyr lock is formed, catalysis only depends on the conserved catalytic residues within the active site. The results of the glutaminase activity assays are now

presented in Fig. 5g in the revised manuscript and described beginning on line 311. A description of the methods used to perform these assays begins on line 580 of the revised manuscript.

Figure 1: cryo-EM analysis of the GLS2 N308A mutant **(a)** 2D classes of the GLS2 N308A mutant indicates it forms a tetramer **(b)** cryo-EM micrograph of the GLS2 N308A mutant in the presence of substrate (glutamine) and Pi (activator) shows no evidence of filament formation.

Figure 2: Substrate dependent glutaminase activity assay (n=3) of GLS2 wild type (red) and GLS2 N308A (blue). X-axis: the concentration of substrate (mM); Y-axis: the initial velocity of enzyme (mM/min). K_m : Michaelis constant (mM); K_{cat} : maximum velocity; n: Hill coefficient. Circles indicate the mean value, and error bars are shown for the standard error. The curve is fit to the Michaelis–Menten equation.

I have a few minor questions and suggestions about how the cryo-EM analysis was performed and is presented, as detailed below, but nothing that calls into question the major findings of the paper.

1) Ext. Data Fig. 5 - The density shown for the "terminal subunit" in the GAC and GLS2 filaments (right panels). It's not clear to me what "terminal subunit" means in this context. Is this just the last tetramer in the reconstruction? If so, then any differences seen between these subunits and subunits near the core would only be the result of heterogeneity in the dataset (likely due to flexibility of the filaments), as tetramers in that position would still be physically in the middle of a filament. If this is the case, then the rationale with regard to the relative ordering of the "terminal" subunit is not sound here, and should be omitted (but, the case for the conformational differences made in the right panel remains sound).

On the other hand, if efforts were made to identify physical ends of filaments from within micrographs and reconstruct those separately, this process should be described in results and methods and the map for this reconstruction should be described in the data table and deposited to EMDB.

Indeed, we used "terminal subunit" to refer to the last tetramer in the reconstruction, but as the reviewer points out, these subunits would still be physically in the middle of the filament. We agree with the reviewer that our original terminology may be misleading, so we have revised the wording as follows: "The missing density of the lid loop of the terminal tetramer in the cryo-EM reconstruction for the constitutively active GLS2 (K253A) filament indicates its flexibility, representing a snapshot of this detachment (Extended Data Fig. 5b, right panel, in the revised manuscript) that occurs following catalytic turnover and product release. We interpret this missing density as heterogeneity in the tetramers of the GLS2 filament, which arises as differences in the terminal tetramers of the cryo-EM reconstruction. However, in the case of the GAC (Y466W) mutant, which binds Pi and substrate but cannot undergo catalysis, the filament persists (Fig. 2e in the revised manuscript) and now the density of the lid loop within the terminal tetramer of the cryo-EM reconstruction remains intact (Extended Data Fig. 5a, right panel)". This is stated in lines 277-285 of the revised manuscript.

2) Image processing and modeling with respect to the ankyrin repeat domains of GLS2. For both the free and filament forms of GLS2, the ankyrin repeat domains appear very poorly ordered in the cryo-EM maps (in the extended figures). It would be helpful to provide images of how the models fit into the maps in this region – even low pass filtered maps that provided some confidence that the overall position is correct would be useful.

We thank the reviewer for the helpful suggestion. To address this, we made a gaussian low pass filtered map for both apo-GLS2 and the GLS2 filament (shown in Fig. 3 below). For apo-GLS2, the low pass filtered map shows strong cryo-EM map density for the ankyrin repeats and the deposited model is in excellent agreement with the cryo-EM map. However, upon inspecting the low pass filtered cryo-EM map for the GLS2 filament, we found that the ankyrin repeats of the GLS2 filament model are a poor fit for the map. In general, this region of the cryo-EM map is not well-resolved and there is not sufficient map density to accurately build an atomic model of the ankyrin repeats, likely due to increased flexibility in this region. Based on our "scissor movement"

model, the N-terminal regions of a GLS2 dimer are further apart in the filament structure in comparison to the apo-GLS2 tetramer, allowing greater movement of the ankyrin repeats which results in weakened cryo-EM map density. While we originally included the ankyrin repeats in the GLS2 filament as observed in the apo-GLS2 structure, after further consideration based on the reviewer's suggestion, we have decided to remove the ankyrin repeats in the deposited GLS2 filament structure. We have updated the revised manuscript accordingly and the new atomic model of GLS2 K253A filament without the ankyrin repeats has been deposited to the PDB. A corresponding validation report is attached (separate file) to this rebuttal letter.

Figure 3. Low pass filtered maps show the fitting of atomic models of **(a)** apo-GLS2, and **(b)** GLS2 K253A filament. The map of Apo-GLS2 is constructed in C2 symmetry and the map of GLS2 K253A filament is constructed in C1 symmetry. The ankyrin repeats are shown in orange.

One concern I have is with the symmetry applied in the reconstructions. Is there sufficient evidence that free GLS2 tetramers actually have C2 symmetry with the ankyrin repeats asymmetrically positioned within dimers? If the ankyrin repeats in the two halves of the tetramer were moving independently, then it should be possible to classify out different states. Similarly for the filament reconstructions, where now the ankyrin repeats appear to be on the same side in each tetramer, a more careful treatment of heterogeneity could provide greater confidence on the position of these domains.

Based on our initial C1 reconstructions (~ 3.4 Å, the low-pass filter map shown below in Fig. 4) of apo-GLS2, we did not observe any clear differences between the two dimers that make up the apo-GLS2 tetramer, and the ankyrin repeats were found to be positioned asymmetrically in the dimers. We attempted extensive 3D classifications to resolve discrete conformations of the ankyrin repeats but were unable to obtain a class with the ankyrin repeats on the opposite side. In addition, the previously solved crystal structure of kidney-type glutaminase (KGA/GLS) shows a similar asymmetric positioning of the ankyrin repeats (Pasquali et al., *J Biol Chem*, 2017). Given the improvement in the map density of the ankyrin repeat regions when C2 is applied, and the agreement with the crystal structure of KGA, we feel that the use of C2 is reasonable in this case. We also corrected our discussion of the positioning of the ankyrin repeats within the apo-GLS2 tetramers (lines 324-329 in the revised manuscript) and expanded the figure legend of Extended Data Figure 7 (lines 928-933).

For the GLS2 filament, we have removed the ankyrin repeats from the deposited atomic model due to weak cryo-EM map density in this region, which we attribute to increased flexibility. In contrast to our apo-GLS2 C1 reconstructions, we did observe differences in the cryo-EM map density of the ankyrin repeats between dimers in the C1 reconstruction of the GLS2 filament (Fig. 3b, above). Because of these observed differences, we opted not to apply symmetry in order to preserve the asymmetric information.

3) *The description of symmetry imposition in general is not very clear. At what steps, for example, were helical symmetry applied in the reconstructions as implied in the flowcharts for the filament reconstructions? Or were helical restraints imposed during refinement but helical symmetry not imposed? Why was point group symmetry not applied to the GLS2 filament reconstruction which, if I understand correctly, has a 2-fold symmetry axis coincident with the helical axis? For the GAC filament, why was C2 symmetry applied (and in which orientation)? If I understand correctly, the GAC filament should actually have D2 symmetry with 2-fold axes coincident with the helical axis and perpendicular to the axis. I think that these issues can generally be resolved with a more careful description of the approach taken and the rationale in the methods section.*

Based on this reviewer's comments, we have tried to provide a clearer description of the data processing. The 'Methods' section has been updated in the revised manuscript (see below; also, lines 641-684 in the revised manuscript). We first used single particle analysis methods to reconstruct the filament, then manually estimated the helical parameters of the filament and used helical reconstruction in cryoSPARC to apply helical symmetry. As discussed above, we did not apply point group symmetry to the GLS2 filament since the cryo-EM map densities of the ankyrin repeat regions between the two dimers were observed to be different, thus, we maintained C1 to avoid losing this apparent asymmetry. However, we agree with the reviewer that the GAC filament could have D2 symmetry. In our original reconstruction of the GAC filament, C2 was applied along the helical axis. Based on the reviewer's suggestion, we performed the reconstruction with D2 symmetry applied. When comparing the cryo-EM maps of the C2 and D2 reconstructions, we found that the maps are nearly identical with only a subtle difference in resolution (C2 3.35 Å vs. D2 3.31 Å). Given the high degree of similarity of the C2 and D2 maps, we decided to keep the C2 map that we have deposited to the wwPDB.

We have provided a more detailed description of our methods/approach as follows:

Cryo-EM data processing

For the apo-GLS2 map, dose-fractionated image stacks were subjected to motioncor2 wrapped in RELION 4.0.0.⁴⁶⁻⁴⁸ through SBGrid, followed by patch CTF estimation in cryoSPARC V4.0.0.⁴⁹. The motion-corrected micrographs with CTF better than 4 Å were passed for later processing. A blob picker in cryoSPARC was used to generate the templates for further Topaz picking⁵⁰, and ~1M particles were picked. The particles were subjected to 3D classification by using *ab initio* reconstruction and heterogenous refinement with four classes, and only the best classes were kept. The remaining ~329K particles were subjected to homogenous and heterogenous refinement. After several rounds of refinement, CTF refinement was carried out for per-particles and per-exposure groups. The final reconstruction with C2 symmetry yields a map with a resolution of 3.12 Å

determined by FSC. The workflow is provided in Extended Data Fig. 8. Cryo-EM maps are visualized in ChimeraX⁵¹.

For the GAC (Y466W) filament structure, dose-fractionated image stacks were subjected to patch motion correction, followed by patch CTF estimation in cryoSPARC V4.2.1. The motion-corrected micrographs with CTF better than 4 Å were passed for later processing. The template-free filament trace job in cryoSPARC picks ~397K particles for 2D classification. About 180K particles from the best 2D classes were extracted and subjected to 3D classification by using *ab initio* reconstruction and heterogenous refinement with four classes. The best (~112K) particles from 3D classification were picked and subjected to multiple rounds of homogenous and non-uniform refinement with CTF refinement for per-particles and per-exposure group, which yielded a 4 Å map. An atomic model of GAC tetramer was docked into the map to measure the helical parameters for further helical reconstruction. Then using helical reconstruction in cryoSPARC, the helical twist was searched between 40-60 degrees and the helical rise was searched between 55-85 Å, which yielded a 3.6 Å map. Another round of 3D classification was performed to further select for the best particles. About 110k particles were selected to undergo multiple rounds of helical refinement with D2 symmetry and CTF refinement and yielded a 3.3 Å map determined by FSC. The optimized helical parameters estimated by cryoSPARC measures a 51° twist and a rise of 68 Å. Finally, the map was sharpened in cryoSPARC. The workflow is provided in Extended Data Fig 9.

For the GLS2 (K253A) filament structure, dose-fractionated image stacks were subjected to patch motion correction, followed by patch CTF estimation in cryoSPARC V4.0.0. The motion-corrected micrographs with CTF better than 4 Å were passed for later processing, which yielded 3,640 micrographs. Manual picking in cryoSPARC was used on 40 micrographs to pick ~600 particles manually, which yielded five classes of templates by 2D classification. The selected template was used for the filament trace job in cryoSPARC to template pick particles. 333K particles were extracted and subjected to 2D classification. 92k particles from the best 2D classes were extracted and subjected to 3D classification by using *ab initio* reconstruction and heterogenous refinement with four classes. The best class with ~48K particles was subjected to multiple rounds of homogenous and non-uniform refinement with CTF refinement for per-particles and per-exposure group. Then using helical reconstruction with C1 symmetry in cryoSPARC, the helical twist was searched between 40-60 degrees and the helical rise was searched between 55-85 Å, which yielded a 3.26 Å map determined by FSC. The optimized helical parameters estimated by cryoSPARC measures a 48° twist and a rise of 66 Å. Finally, the quality of the map was improved by Phenix using density modification and local sharpening. The workflow is provided in Extended Data Fig 10.

I would also suggest, given the recent body of work describing mechanisms by which other enzymes are regulated by polymerization into filaments, that the authors include a brief discussion of how their results fit into this field, comparing how polymerization affects glutaminase activity with how it appears to affect activity and regulation in other enzymes. Of course, this is merely a suggestion that might help to broaden the appeal beyond the audience that will be interested in glutaminase regulation specifically.

We thank the reviewer for the suggestion and have added the text below to the revised manuscript (lines 377-387):

Previously, it has been reported that various metabolic enzymes assemble into filamentous structures (1, 2). Several recent structural studies leveraging the advances in cryo-EM have demonstrated that filament assembly allows for allosteric regulation of enzymes (3-6). Our structures of glutaminase filaments demonstrate that filament formation stabilizes the allosteric sites on glutaminase (i.e., the activation loop) in a manner similar to that observed in phosphoribosyl pyrophosphate synthetase 1 (PRPS1) (5). In the case of PRPS1, filament formation enabled binding of the activator, Pi. In contrast, we find that Pi, or the introduction of mutations that result in a constitutively active enzyme, is required for filament formation. Given that the presence of substrate is also required for glutaminase filament formation, we propose that filament formation stabilizes the active form of glutaminase, similar to what has been suggested for the enzyme CTP synthase 1 (4).

Reviewer #2:

In this study, Feng et al. have taken an integrated, multidisciplinary approach incorporating structural, biochemical, and biophysical tools to unravel the molecular mechanisms of the GAC and GSL2 enzymes. To begin exploring the composition, architecture, and functionality of these proteins, the authors first demonstrated the ability of these glutaminase enzymes to form high-order assemblies and determined the cryoEM structures of GAC and GSL2 filaments at 3.35 and 3.26 angstrom resolutions, respectively. Within these assemblies, the basic building block of the helical structure is comprised of a tetramer. This unit then interacts with neighboring tetramers, which results in helical stacking of GAC and GSL2 tetramers and elongation of filaments. The authors further show that these filaments contain novel structural features that are critical for polymerization. Similar to other metabolic and filamentous enzyme structures, GAC and GSL2 robustly polymerize into filaments to exert catalytic activity. In addition to modulating enzyme activity through polymerization, an unexpected feature of these filaments was the involvement of ankyrin domains in the regulation of enzyme activity. In summary, this study provides novel insight into the mechanism of polymerization of GAC and GSL2 enzymes and the regulation of their catalytic activity.

We thank the reviewer for the careful consideration and general appreciation of our study.

More detailed comments:

- 1. Between lines 55 and 57, the authors claim that the importance of higher-order structures of enzyme regulation and function has not been definitively established. However, there are many published studies (especially within the past few years) describing the regulation of enzyme activity by filament formation. The authors should introduce or discuss these studies in their manuscript and broaden their analysis of filamentous enzymes.*
- 2. How do their insights into the enzyme assembly and regulation of catalytic activity by filament formation compare with other published studies?*

This point was also raised by reviewer 1. As described above, we have removed the lines 55-57 and added a paragraph to the revised manuscript (lines 377-387) that describes other findings of filament formation in metabolic enzymes and how they compare to what we describe in this study.

3. To validate their structural models, the authors should characterize the residues located at the oligomerization interfaces by mutagenesis and functional studies and discuss the molecular mechanism of filament assembly.

Here again, this point was also raised by Reviewer #1 and is addressed above. We mutated N308 on the GLS2 filament interface (N308A) to interfere with filament formation (Fig. 1, above) and performed glutaminase activity assays to determine the kinetic parameters impacted by filament formation. As shown in Fig. 2, above, we found that the K_m value for the N308A mutant increased significantly compared to the K_m determined for the wild-type enzyme, while K_{cat} appears to be unaffected. In addition, two recently published studies on GLS1 filaments (Guo et al., *Cell Research*, 2023; Adamoski et al., *Nat Struct Mol Biol*, 2023) demonstrated that a single mutation on the filament interface of GAC decreases enzymatic activity, which matches our findings in GLS2. We now include the activity assay results (Fig. 5g) in the revised manuscript and we describe the results on line 309. Please also find a description the detailed catalytic mechanism in the subsection of the “Results”, “A mechanism for glutaminase activation coupled to filament formation” (starting on line 255 of the revised manuscript), and starting on line 388 under Discussion section.

4. More detailed description of cryoEM data processing strategy is needed in the methods section. Did the authors use local refinement, CTF refinement, or any other strategies to process their micrographs? If yes, the data processing scheme should be discussed in the methods section.

5. How were the helical parameters determined? Did the authors use cryoEM helical reconstruction tools? The details should be included in the methods section.

In response to the reviewer’s points 4 and 5, we have updated the ‘Methods’ section (starting on line 640) in the revised manuscript to include this information (also as described in our response to Reviewer 1). Briefly, we used the helical reconstruction tools in cryoSPARC to determine the helical parameters. Specifically, we measured the spacing between the repeating units of filament on the micrographs, to estimate the required box size. Then, we extracted the particles and ran a round of *ab-initio* reconstruction with multiple classes to pick the best class. Using the *ab-initio* map, we performed multiple rounds of homogenous and non-uniform refinement to improve the resolution. At this stage, an initial atomic model was fit into the map to enable manual measurement of the helical parameters (i.e., twist and rise). Then, we performed multiple rounds of helical refinement using the estimated helical parameters as a starting point to improve the resolution of the map. Before each round of refinement, we performed CTF refinement on both per-particles and per-exposure group, which improved the resolution of the reconstruction. We also tried local refinement on the ankyrin repeats region of both apo-GLS2 and GLS2 filaments, but we were not able to improve the resolution of the region.

6. All three models have high clash scores for the reported resolution range. The authors should re-refine their structures and fix these errors. The rotamer and Ramachandran outliers should also be fixed and the validation reports should be updated.

Based on the reviewer's comments, we attempted to resolve this using multiple rounds of Phenix real space refinement as well as manual adjustments in Coot. However, given the large interface area not only between tetrameric units of the filament, but also between the monomers within the tetramers, we were unable to achieve lower clash scores. In addition, we have only one Ramachandran outlier in the apo-GLS2 structures and just two Ramachandran outliers per tetramer in the GAC and GLS2 filament structures. We attempted to fix these residues but ended up with worse results, since the outliers are found in flexible regions and adjustments made to these residues lead to poorer statistics for adjacent residues. Overall, we feel that the quality of our structures is on par with similar cryo-EM structures, and all the key residues described in the manuscript are well resolved and support our conclusions.

Minor comment:

1. Typo: Thermo Fisher Scientific was misspelled throughout the methods section.

We thank the reviewer for identifying this mistake, which has been updated in the revised manuscript.

REFERENCES added to the revised manuscript starting on line 377.

28. C. K. Park, N. C. Horton, Structures, functions, and mechanisms of filament forming enzymes: a renaissance of enzyme filamentation. *Biophysical reviews* **11**, 927-994 (2019).
43. K. L. Hvorecny, J. M. Kollman, Greater than the sum of parts: Mechanisms of metabolic regulation by enzyme filaments. *Current Opinion in Structural Biology* **79**, 102530 (2023).
44. A. L. Burrell *et al.*, IMPDH1 retinal variants control filament architecture to tune allosteric regulation. *Nature Structural & Molecular Biology* **29**, 47-58 (2022).
45. J. M. Hansen *et al.*, Cryo-EM structures of CTP synthase filaments reveal mechanism of pH-sensitive assembly during budding yeast starvation. *eLife* **10**, e73368 (2021).
46. K. L. Hvorecny, K. Hargett, J. D. Quispe, J. M. Kollman, Human PRPS1 filaments stabilize allosteric sites to regulate activity. *Nature Structural & Molecular Biology* **30**, 391-402 (2023).
47. M. Zhang *et al.*, Structural basis for the catalytic activity of filamentous human serine beta-lactamase-like protein LACTB. *Structure* **30**, 685-696.e685 (2022).

REVIEWER COMMENTS

Reviewer #1 (Remarks to the Author):

The authors have nicely addressed my major concerns, and all but one of my minor concerns. My only remaining concern is about interpretation of differences between terminal subunits in the reconstructions (my initial minor concern #1).

If I understand correctly, the authors see differences in the lid loop occupancy in the terminal densities for GLS2(K253A) and GAC(Y466W), which they interpret as indicating greater heterogeneity of the lid loop in GLS2(K253A). However, I assume that they are focused on the differences at the ends of the reconstruction because they don't see similar differences of the lid loops at the center of the reconstruction. If that's not correct, then there is something wrong with my understanding that hopefully the authors can clarify. But if it is correct that the central lid loops have similar occupancy then a problem arises with their interpretation.

If overlapping segments of the filaments are extracted from the micrographs, then individual protomers are averaged in the reconstruction multiple times in different positions. So, the same tetramer may be included in the reconstruction in the i , $i+1$, $i+2$, etc. positions. It therefore doesn't make sense to me to interpret differences at the ends of the filament as demonstrating heterogeneity of specific regions that do not appear to be heterogeneous in the center of the reconstruction.

Instead, I think the most straightforward interpretation would be about the rigidity of helical symmetry between GAC and GLS2. In a more flexible filament that deviates more from perfect helical symmetry, one would expect greater alignment error as you move from the center of the filament outward, which would give rise to greater disorder the further one gets from the center of the reconstruction. Although local resolution maps are not presented, the overall length of the ordered part of the filaments shown in extended data figures 9 and 10 suggests this is indeed the case for GLS2 being overall more flexible than GAC.

I guess another way to put it is that I expect that the overall quality of the map for terminal subunits is worse for GLS2 than for GAC, rather than this being a specific feature of the lid loop occupancy.

This is a fairly minor technical point, and should not hold up acceptance of the paper, but I think it's important to discuss this heterogeneity accurately. Again, if I have completely misunderstood the situation I would welcome further clarification from the authors.

Reviewer #2 (Remarks to the Author):

The authors have adequately addressed most of my previous comments. The following papers should also be cited in the manuscript: PMID: 29899443, PMID: 28646105, PMID: 36534696, PMID: 31999252. The only remaining item is that they must correct the chemical errors in their models. All three models still have very high clash scores and rotamer and Ramachandran outliers. The authors argue that the quality of these models is on par with similar cryoEM structures but the models published in PMID: 36534696, PMID: 35013599, PMID: 31586059, PMID: 33875834, and others have significantly lower clash scores and almost no rotamer and Ramachandran outliers. The authors could utilize phenix real space refinement and Coot to fix these issues and/or stub the side chain outliers. Once these models are corrected, I would recommend the publication of the manuscript.

We sincerely thank the reviewers for their insightful comments and general support of our study. We have revised the manuscript to address their concerns in the second round of revision. Our point-by-point response to each of the reviewers' comments is provided below.

Reviewer #1 (Remarks to the Author):

The authors have nicely addressed my major concerns, and all but one of my minor concerns. My only remaining concern is about interpretation of differences between terminal subunits in the reconstructions (my initial minor concern #1).

If I understand correctly, the authors see differences in the lid loop occupancy in the terminal densities for GLS2(K253A) and GAC(Y466W), which they interpret as indicating greater heterogeneity of the lid loop in GLS2(K253A). However, I assume that they are focused on the differences at the ends of the reconstruction because they don't see similar differences of the lid loops at the center of the reconstruction. If that's not correct, then there is something wrong with my understanding that hopefully the authors can clarify. But if it is correct that the central lid loops have similar occupancy then a problem arises with their interpretation.

If overlapping segments of the filaments are extracted from the micrographs, then individual protomers are averaged in the reconstruction multiple times in different positions. So, the same tetramer may be included in the reconstruction in the i , $i+1$, $i+2$, etc. positions. It therefore doesn't make sense to me to interpret differences at the ends of the filament as demonstrating heterogeneity of specific regions that do not appear to be heterogeneous in the center of the reconstruction.

Instead, I think the most straightforward interpretation would be about the rigidity of helical symmetry between GAC and GLS2. In a more flexible filament that deviates more from perfect helical symmetry, one would expect greater alignment error as you move from the center of the filament outward, which would give rise to greater disorder the further one gets from the center of the reconstruction. Although local resolution maps are not presented, the overall length of the ordered part of the filaments shown in extended data figures 9 and 10 suggests this is indeed the case for GLS2 being overall more flexible than GAC.

I guess another way to put it is that I expect that the overall quality of the map for terminal subunits is worse for GLS2 than for GAC, rather than this being a specific feature of the lid loop occupancy.

This is a fairly minor technical point, and should not hold up acceptance of the paper, but I think it's important to discuss this heterogeneity accurately. Again, if I have completely misunderstood the situation I would welcome further clarification from the authors.

As this reviewer points out, the difference of lid loop occupancy may be contributed by a greater alignment error of GLS2 filament reconstruction, which suggests that the GLS2 filaments may be more flexible than GAC filaments. As suggested by the reviewer, we generated and compared the local resolution map of both the GAC and GLS2 filaments. Indeed, the local

resolution of GLS2 filament gradually decays moving from the center to the edge of the reconstruction, while GAC has a more uniform distribution of the local resolution (see figure 1 below). We have added the text below to the revised the manuscript (see lines 277-288).

“The missing density of the lid loop of the terminal tetramer in the cryo-EM reconstruction for the constitutively active GLS2 (K253A) filament indicates its flexibility, representing a snapshot of this detachment (Extended Data Fig. 5b, right panel) that occurs following catalytic turnover and product release. However, in the case of the GAC (Y466W) mutant, which binds Pi and substrate but cannot undergo catalysis, the filament persists (Fig. 2e) and now the density of the lid loop within the terminal tetramer of the cryo-EM reconstruction remains intact (Extended Data Fig. 5a, right panel). We interpret the different occupancy of the lid loop density as the rigidity of helical symmetry between GAC and GLS2. In the more flexible GLS2 filament, the alignment error is greater at the distal regions of the cryo-EM map, resulting in lower local resolution for the terminal tetramers, thus causing weaker electron density occupancy of the lid loop in the reconstruction, while the local resolution of the GAC Y466W filament is more consistent across the whole map (Extended Data Figs. 9, 10).”

Reviewer #2 (Remarks to the Author):

The authors have adequately addressed most of my previous comments. The following papers should also be cited in the manuscript: PMID: 29899443, PMID: 28646105, PMID: 36534696, PMID: 31999252. The only remaining item is that they must correct the chemical errors in their models. All three models still have very high clash scores and rotamer and Ramachandran outliers. The authors argue that the quality of these models is on par with similar cryoEM structures but the models published in PMID: 36534696, PMID: 35013599, PMID: 31586059, PMID: 33875834, and others have significantly lower clash scores and almost no rotamer and Ramachandran outliers. The authors could utilize phenix real space refinement and Coot to fix these issues and/or stub the side chain outliers. Once these models are corrected, I would recommend the publication of the manuscript.

To address one of the remaining concerns of this reviewer, we have added the citations requested in the text below (see lines 380-382 in the revised manuscript, citations 48-51).

“Previously, it has been reported that various metabolic enzymes assemble into filamentous structures^{28,43}. Several recent structural studies leveraging the advances in cryo-EM have demonstrated that filament assembly allows for allosteric regulation of enzymes⁴⁴⁻⁵¹.”

In addition, we have corrected our atomic models using phenix real space refinement and Coot and successfully fixed the issues of all three models. Now all the models have Clashscore of 5 with 0% Ramachandran outliers, and very few rotamer outliers (please see the updated validation reports and figure 2 below). We compared the corrected models with the original models, and confirmed that all the key residues discussed in the manuscript have not changed from their original positions. We also updated Extended data table 1 to reflect the corrected models.

a

GAC Y466W filament

Local resolution (Å)

b

GLS2 K253A filament

Local resolution (Å)

Figure 1. Local resolution maps of **(a)** GAC Y466W filament and **(b)** GLS2 K253A filament.

Apo GLS2

GAC Y466W filament

GLS2 K253A filament

Figure 2. Validation metrics of structures reported in this manuscript.

Reference:

48. J. A. Bennett, L. R. Steward, J. Rudolph, A. P. Voss, H. Aydin, The structure of the human LACTB filament reveals the mechanisms of assembly and membrane binding. *PLoS Biol* **20**, e3001899 (2022).
49. M. Hunkeler *et al.*, Structural basis for regulation of human acetyl-CoA carboxylase. *Nature* **558**, 470-474 (2018).
50. M. C. Johnson, J. M. Kollman, Cryo-EM structures demonstrate human IMPDH2 filament assembly tunes allosteric regulation. *Elife* **9** (2020).
51. B. A. Webb, A. M. Dosey, T. Wittmann, J. M. Kollman, D. L. Barber, The glycolytic enzyme phosphofructokinase-1 assembles into filaments. *J Cell Biol* **216**, 2305-2313 (2017).